## JCB Journal of Cell Biology

# SGK regulates pH increase and cyclin B–Cdk1 activation to resume meiosis in starfish ovarian oocytes

Enako Hosoda[1], Daisaku Hiraoka[2], Noritaka Hirohashi[1], Saki Omi[1], Takeo Kishimoto[2], and Kazuyoshi Chiba[1]

**Tight regulation of intracellular pH ($pH_i$) is essential for biological processes. Fully grown oocytes, having a large nucleus called the germinal vesicle, arrest at meiotic prophase I. Upon hormonal stimulus, oocytes resume meiosis to become fertilizable. At this time, the $pH_i$ increases via $Na^+/H^+$ exchanger activity, although the regulation and function of this change remain obscure. Here, we show that in starfish oocytes, serum- and glucocorticoid-regulated kinase (SGK) is activated via PI3K/TORC2/PDK1 signaling after hormonal stimulus and that SGK is required for this $pH_i$ increase and cyclin B–Cdk1 activation. When we clamped the $pH_i$ at 6.7, corresponding to the $pH_i$ of unstimulated ovarian oocytes, hormonal stimulation induced cyclin B–Cdk1 activation; thereafter, oocytes failed in actin-dependent chromosome transport and spindle assembly after germinal vesicle breakdown. Thus, this SGK-dependent $pH_i$ increase is likely a prerequisite for these events in ovarian oocytes. We propose a model that SGK drives meiotic resumption via concomitant regulation of the $pH_i$ and cell cycle machinery.**

## Introduction

Intracellular pH ($pH_i$) is tightly regulated in living cells. Increases in $pH_i$ are required for physiological and pathological processes, including early embryonic development (Schatten et al., 1986; Grandin and Charbonneau, 1990; Baltz, 1993) and cancer cell survival (Grillo-Hill et al., 2015). A key $pH_i$ regulator, the sodium–proton exchanger (NHE), is a 12-transmembrane protein that increases $pH_i$ by exporting intracellular $H^+$ and importing extracellular $Na^+$ (Orlowski and Grinstein, 2004). Many kinases, including serum- and glucocorticoid-regulated kinase (SGK; Yun, 2003; Wang et al., 2005; He et al., 2011) and p90RSK (Takahashi et al., 1999), activate NHE in mammalian cells.

An NHE-dependent $pH_i$ increase has also been reported during oocyte maturation (Miyazaki et al., 1975; Lee and Steinhardt, 1981; Rezai et al., 1994; Dubé and Eckberg, 1997; Harada et al., 2003; Moriwaki et al., 2013), an essential process to generate fertilizable eggs. The underlying regulatory mechanism and function of this $pH_i$ increase are not completely understood. In the ovaries of most animals, fully grown oocytes, which have a large nucleus called the germinal vesicle (GV), are arrested at prophase of meiosis I (ProI; Masui, 2001). Upon extracellular stimulation by maturation-inducing hormones, ovarian oocytes resume meiosis via activation of cyclin B–Cdk1 and subsequently undergo GV breakdown (GVBD; Hunt, 1989; Nurse, 1990; Kishimoto, 2015). Subsequently, oocytes undergo a

secondary arrest at the metaphase of meiosis I or II (MI or MII arrest) that enables successful fertilization (Chiba, 2011). Starfish oocytes have long been used as a model to study oocyte maturation (Kishimoto, 2018). Previously, we reported that $pH_i$ of ProI-arrested ovarian oocytes is low (∼6.7) due to the relatively high $CO_2$ and low $O_2$ concentrations in the body cavity of starfish (Moriwaki et al., 2013). Soon after stimulation by the hormone 1-methyladenine (1-MA; Kanatani et al., 1969), $pH_i$ increases to ∼6.9 (Moriwaki et al., 2013), after which the oocytes undergo GVBD and ultimately arrest at MI (Harada et al., 2003; Moriwaki et al., 2013). To maintain the MI arrest, $pH_i$ is maintained at ∼6.9 until spawning (Harada et al., 2003, 2010; Oita et al., 2004; Usui et al., 2008; Moriwaki et al., 2013; Ochi et al., 2016). The $pH_i$ increase induced by 1-MA requires starfish NHE3 (sfNHE3; Harada et al., 2003, 2010; Moriwaki et al., 2013). However, the upstream signaling and function of this $pH_i$ increase remain elusive.

For analysis of $pH_i$ regulation, unstimulated starfish oocytes are isolated from ovaries and placed in artificial seawater (ASW), after which various techniques, such as microinjection, can be easily performed. ASW contains relatively low $CO_2$ and high $O_2$ relative to the inner body cavity (Moriwaki et al., 2013). Due to this difference in gas concentrations, the basal $pH_i$ of isolated oocytes is ∼7.0, ∼0.3 units higher than that in ovarian

[1]Department of Biological Sciences, Ochanomizu University, Tokyo, Japan;    [2]Science and Education Center, Ochanomizu University, Tokyo, Japan.

Correspondence to Kazuyoshi Chiba: chiba.kazuyoshi@ocha.ac.jp;    N. Hirohashi's present address is Oki Marine Biological Station, Education and Research Center for Biological Resources, Shimane, Japan.



oocytes (Moriwaki et al., 2013). After 1-MA treatment, this value increases further, to ~7.3, in a sfNHE3-dependent manner (Harada et al., 2003, 2010; Moriwaki et al., 2013). 1-MA induces Gβγ-dependent activation of phosphoinositide 3-kinase (PI3K; Shilling et al., 1989; Chiba et al., 1993; Sadler and Ruderman, 1998; Vanhaesebroeck et al., 2010; Hiraoka et al., 2016), and we demonstrated that the PI3K pathway mediates the pH$_i$ increase (Harada et al., 2003). The molecular link between PI3K and the sfNHE3-dependent pH$_i$ increase is a key issue to be resolved.

One candidate for this link is SGK, which is activated downstream of PI3K and up-regulates NHE3 in mammalian cells (Kobayashi et al., 1999; Tessier and Woodgett, 2006b; He et al., 2011; Malik et al., 2018). Mammalian SGK belongs to the AGC kinase family (Pearce et al., 2010) and has three isoforms: SGK1, 2, and 3 (Kobayashi et al., 1999; Kobayashi and Cohen, 1999; Tessier and Woodgett, 2006a). All three isoforms have an activation loop (A-loop) and a C-terminal hydrophobic motif (HM; Kobayashi et al., 1999), but only SGK3 has the phospholipid-binding domain called the Phox homology domain (PX domain; Tessier and Woodgett, 2006a). Upon agonist stimulation, the HM is first phosphorylated by mammalian target of rapamycin complex 2 (mTORC2) in a PI3K-dependent manner (Kobayashi and Cohen, 1999; Tessier and Woodgett, 2006a; García-Martínez and Alessi, 2008). Phosphoinositide-dependent kinase 1 (PDK1) interacts with the phosphorylated HM and phosphorylates the A-loop (Biondi et al., 2001), resulting in SGK activation (Kobayashi et al., 1999; Lu et al., 2010; Lien et al., 2017; Malik et al., 2018). Although it is not known whether SGK is activated in starfish oocytes, we previously showed that the C terminus of sfNHE3 can be phosphorylated by recombinant human SGK1 in vitro (Harada et al., 2010). Furthermore, PDK1 and TORC2 are functional in starfish oocytes. They activate another AGC family kinase, Akt (Pearce et al., 2010), through phosphorylation of its A-loop and the HM (Hiraoka et al., 2004, 2011). In this context, Akt participates in inhibition of Myt1 and activation of Cdc25 (Okumura et al., 1996, 2002; Hiraoka et al., 2011, 2016), leading to cyclin B–Cdk1 activation through dephosphorylation of Cdk1 at Thr14 and Tyr15 (Okumura et al., 1996; Kishimoto, 2018). Collectively, these observations inspired us to investigate whether SGK serves as a downstream mediator of PI3K for the pH$_i$ increase in starfish oocytes.

Another issue that remains to be resolved is the role of this pH$_i$ increase. Previously, we established a method for clamping pH$_i$ at a desired value in starfish oocytes (Moriwaki et al., 2013), enabling us to examine the effects of altered pH$_i$.

In this study, we found that SGK activation is a prerequisite for two mutually independent events in starfish oocytes: pH$_i$ increase and cyclin B–Cdk1 activation. Moreover, we showed that oocytes with reduced pH$_i$ values exhibit defects in processes toward spindle assembly after GVBD. Based on these findings, we propose a model for SGK-dependent meiotic resumption in starfish ovarian oocytes.

## Results

### Starfish SGK (sfSGK) is a homologue of human SGK3
First, we cloned the cDNA of sfSGK. Its ORF encodes a polypeptide of 489 aa with a predicted molecular mass of 56 kD.

sfSGK has a PX domain and a catalytic domain, whose amino acid sequences are 52% and 77% identical to those of the human SGK3, respectively (Fig. 1 A). There are two possible conserved activating phosphorylation sites in sfSGK: Thr312 in the A-loop and Thr479 in the HM (corresponding to human SGK3 Thr320 and Ser486, respectively; Fig. 1 A). The transcriptome database contains no other SGK isoforms for starfish, suggesting that SGK3 is the only member of this protein family in starfish.

### sfSGK is activated after 1-MA stimulus
Next, we generated two types of antibodies: one against a sfSGK fragment lacking the N-terminal 50 aa (anti-sfSGK-ΔN50) and the other against a sfSGK C-terminal 17-aa peptide containing the HM (anti-sfSGK-HM). Both antibodies detected a protein of 56 kD, corresponding to the predicted molecular mass of sfSGK, in an immunoblot of unstimulated starfish oocytes (Fig. 1 B), suggesting that these antibodies recognize sfSGK. A mobility shift after 1-MA treatment (Fig. 1 B) suggests that sfSGK is phosphorylated after 1-MA stimulation.

Full activation of mammalian SGK is achieved by the A-loop phosphorylation (Kobayashi et al., 1999; Biondi et al., 2001). The amino acid sequence of the A-loop in human SGK3 is similar to that of sfSGK (Fig. 1 A). Hence, to determine whether sfSGK is phosphorylated in the A-loop after 1-MA stimulus, we used a commercial anti-human phospho-SGK antibody that recognizes phosphorylated A-loops in the human SGKs. The antibody detected an ~59 kD protein in 1-MA–stimulated oocytes, but not in unstimulated oocytes (Fig. 1 B). This band ran at the same position as the shifted band of sfSGK (Fig. 1 B). Moreover, when we immunoprecipitated sfSGK using the anti-sfSGK-HM antibody and immunoblotted with the anti-human phospho-SGK antibody, the ~59 kD protein was detected in the bead fraction (Fig. 1 C, upper panel, lane 8) but not in the flow-through fraction (Fig. 1 C, upper panel, lane 5), indicating that this phospho-SGK antibody reacted with sfSGK. Accordingly, hereafter we refer to the A-loop phosphorylation of sfSGK detected by this antibody as "sfSGK-pT312 (A-loop)" in figures. Taken together, these results suggest that sfSGK is phosphorylated at its A-loop, and thereby activated, after 1-MA stimulation.

### sfSGK is activated simultaneously with Akt and before cyclin B–Cdk1 activation
To compare the phosphorylation dynamics of sfSGK with those of Akt, Cdc25, and Cdk1, oocytes were treated with 1-MA in ASW and analyzed by immunoblotting (Fig. 2 A). Mobility shift and A-loop phosphorylation of sfSGK were detected 1 min after 1-MA stimulation. This timing was similar to that of Akt activation (represented by phosphorylation of HM; Fig. 2 A, Akt-pS477 [HM]; Hiraoka et al., 2011). The A-loop phosphorylation level of sfSGK increased rapidly, with a peak at 3 min, followed by a gradual decrease. A band shift of Cdc25 corresponding to hyperphosphorylation (Okumura et al., 1996; Hiraoka et al., 2016) was detected after 7 min (Fig. 2 A). Subsequently, activation of Cdk1 was detectable as dephosphorylation of Tyr15 at 10 min (Fig. 2 A), followed by GVBD (at which the rim of GV becomes fuzzy under differential interference contrast [DIC] microscopic observation; Lénárt et al., 2003) at ~17 min. These results

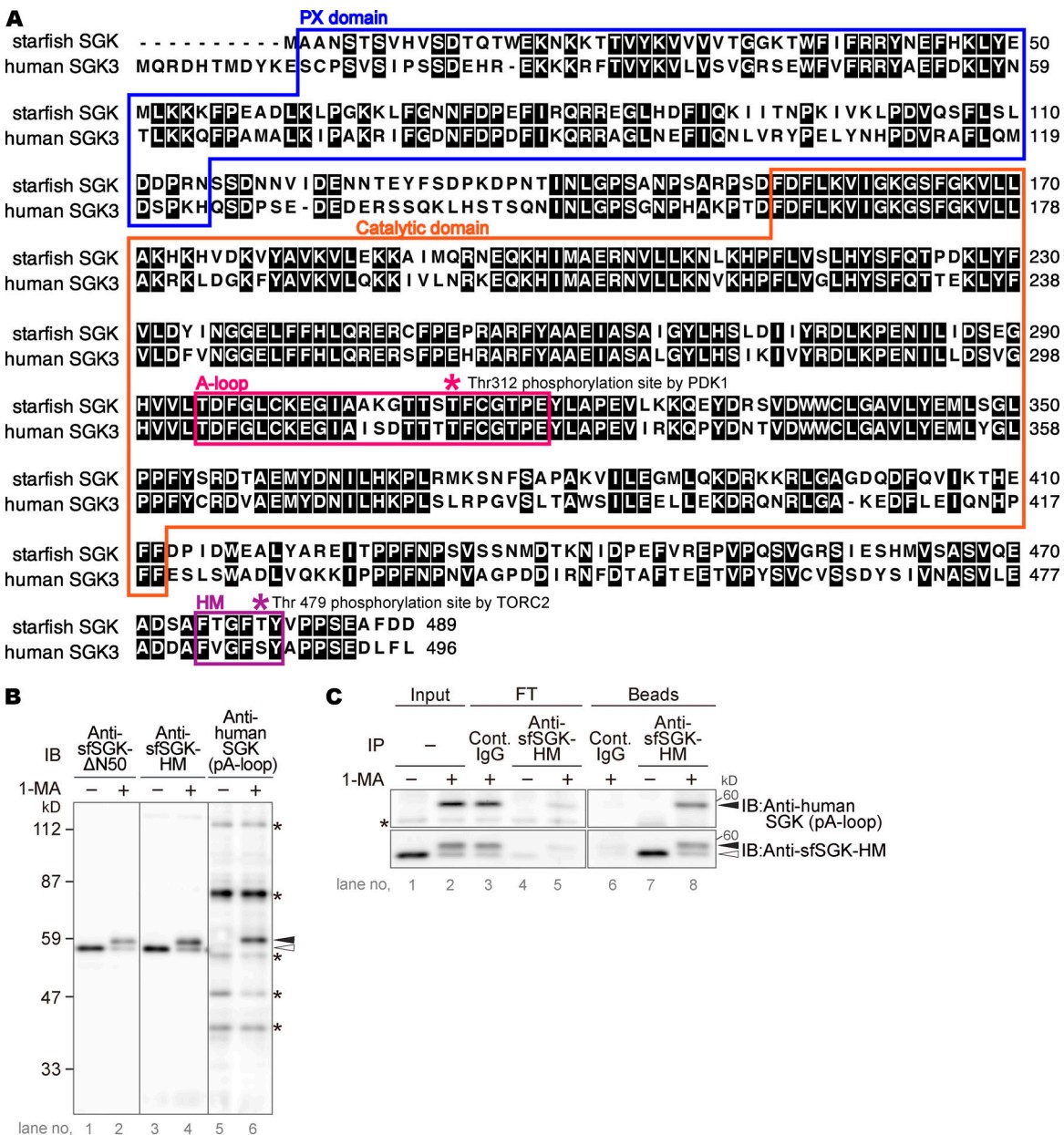

**Figure 1. sfSGK protein and phosphorylation of its A-loop are detectable in starfish oocytes. (A)** The amino acid sequence of sfSGK was aligned with that of human SGK3. Colored boxes indicate conserved domains: PX domain (blue), A-loop (magenta), HM (purple), and catalytic domain (orange). Magenta and purple asterisks indicate conserved residues phosphorylated by PDK1 and TORC2, respectively (see also Fig. 2). **(B)** Unstimulated oocytes were incubated with 1-MA for 4 min, followed by immunoblotting with the indicated antibodies. Asterisks, nonspecific bands. **(C)** Immunoprecipitation (IP) was performed with anti-sfSGK-HM antibody or control (Cont.) IgG using extracts of unstimulated or 1-MA–stimulated oocytes. Input extract (Input), flow-through (FT), and beads (Beads) samples were analyzed by immunoblotting (IB). Asterisk, nonspecific bands. The indicated results in B and C are representative of three and two independent experiments, respectively. Closed and open arrowheads in B and C indicate positions of the upper and lower bands of sfSGK, respectively.

indicate that sfSGK is rapidly activated at the same time as Akt and before Cdc25 hyperphosphorylation and cyclin B–Cdk1 activation.

Next, we investigated phosphorylation of these proteins in ovarian oocytes in the body cavity. To this end, we injected 1-MA into the body cavities of starfish, and then at each time point, isolated two pieces of ovary: one for counting GVBD, and the other for immunoblotting. 95% of ovarian oocytes underwent GVBD within 25 min after 1-MA injection; spawning started at

30 min. For immunoblotting, the piece of ovary was placed directly into sample buffer immediately after isolation. Although these samples contained not only oocytes but also other cells derived from ovarian germinal epithelium, we ignored the involvement of the germinal epithelium because the sfSGK, Cdc25, and Cdk1 proteins were undetectable in it (Fig. 2 B, lane 2). Immunoblotting showed that both sfSGK and Cdc25 in ovarian oocytes were phosphorylated within 5 min and that Tyr15 of Cdk1 was dephosphorylated within 25 min (Fig. 2 C), indicating

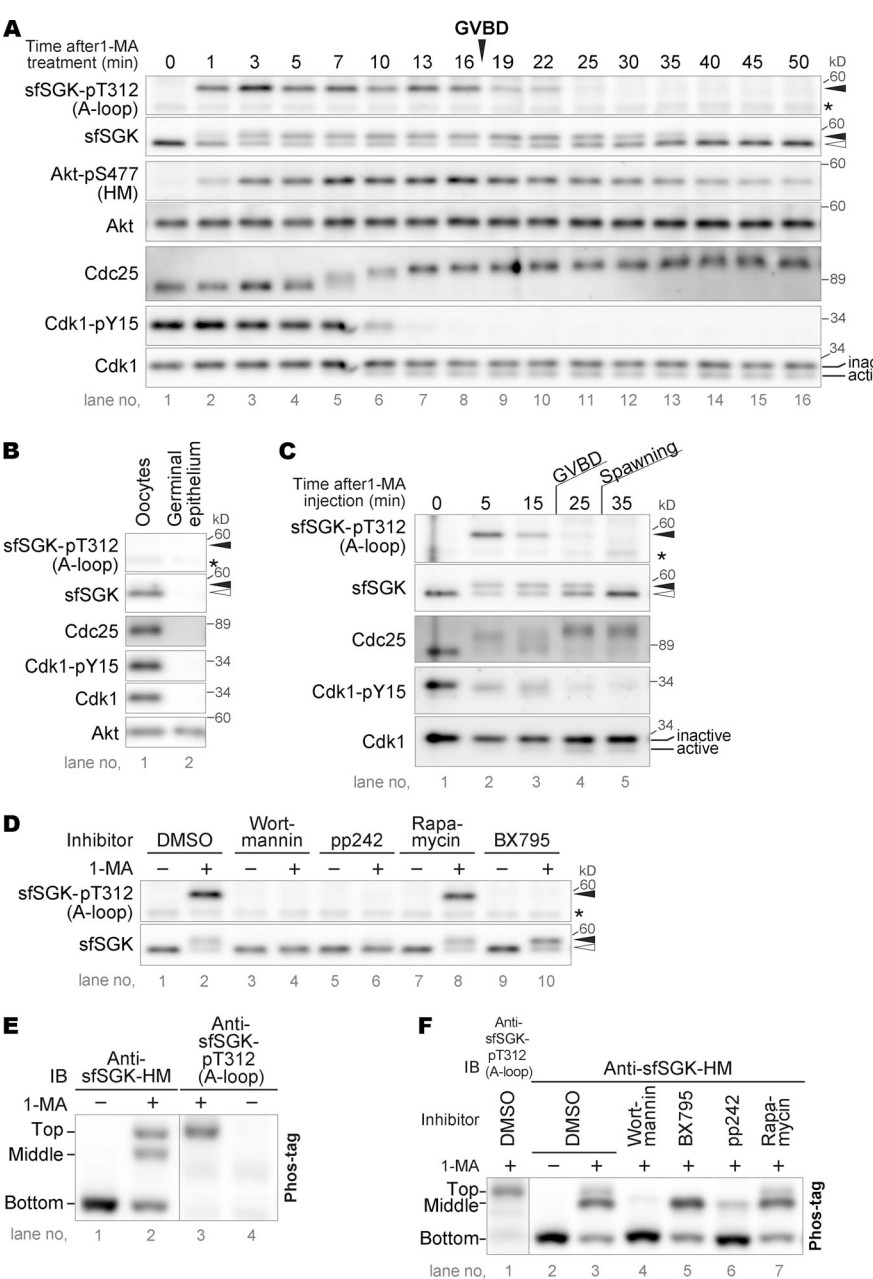

Figure 2. **sfSGK is activated by PDK1 and TORC2 in a PI3K-dependent manner. (A)** Unstimulated oocytes were treated with 1-MA and analyzed by immunoblotting with anti-human phospho-SGK (sfSGK-pT312 [A-loop]), anti-sfSGK-HM (sfSGK), anti-sfAkt phospho-Ser477 (Akt-pS477[HM]), anti-sfAkt C-terminal fragment (Akt), anti-sfCdc25 (Cdc25), anti-Cdk1 phospho-Tyr15 (Cdk1-pY15), and anti-PSTAIR (Cdk1) antibodies. GVBD occurred at 17 min. **(B)** A piece of ovary was separated into oocytes and germinal epithelium, and subjected to immunoblotting. **(C)** To stimulate ovarian oocytes, 1-MA was injected into the body cavity of female starfish. Pieces of ovaries were then recovered at the indicated times and analyzed via immunoblotting or monitored for GVBD timing. GVBD occurred within 25 min. Spawning started at 30 min. **(D)** Unstimulated oocytes were treated with 1-MA for 4 min in the presence of indicated inhibitors or DMSO, and then subjected to normal SDS-PAGE, followed by immunoblotting. Closed and open arrowheads in A–D indicate the positions of the upper and lower sfSGK bands, respectively. Asterisks in A–D, nonspecific bands. **(E and F)** Unstimulated oocytes were treated with 1-MA for 4 min (E) or treated as in D (F), and then subjected to Phos-tag SDS-PAGE, followed by immunoblotting (IB). The results shown are representative of three independent experiments in A, E, and F, and two independent experiments in B–D.

that these proteins were activated in ovarian oocytes similarly to those in isolated oocytes.

It should be noted that the shifted bands of sfSGK were maintained even after disappearance of the A-loop phosphorylation (sfSGK and sfSGK-pT312 [A-loop] at 25–35 min in Fig. 2 A and at 25 min in Fig. 2 C), suggesting the existence of a phosphorylation site outside the A-loop, possibly at the conserved site in the HM.

### PDK1 and TORC2 phosphorylate sfSGK in a PI3K-dependent manner

Next, we investigated the possible involvement of PDK1 and TORC2 in the activation of sfSGK. BX795, a PDK1 inhibitor, completely blocked sfSGK A-loop phosphorylation (Fig. 2 D), suggesting that this phosphorylation depends on PDK1. In

mammalian cells, phosphorylation of the A-loop by PDK1 requires prior phosphorylation of the HM by mammalian TORC2 (Kobayashi and Cohen, 1999; Biondi et al., 2001; Tessier and Woodgett, 2006a; Lu et al., 2010; Lien et al., 2017; Malik et al., 2018). This regulation is likely conserved in starfish because A-loop phosphorylation was also blocked by pp242, a specific inhibitor of target of rapamycin, the catalytic subunit of TORC2 (Fig. 2 D). Notably, the mobility shift of sfSGK was reduced by pp242, but was not affected by BX795 (Fig. 2 D, lanes 6 and 10), suggesting that the shift is caused by TORC2-dependent HM phosphorylation. It should be noted that target of rapamycin forms two complexes: TORC1 and TORC2 (Eltschinger and Loewith, 2016). They are inhibited by pp242. However, we concluded that TORC2, but not TORC1, phosphorylates sfSGK because neither A-loop phosphorylation nor the mobility shift of

sfSGK was blocked by the TORC1 inhibitor rapamycin (Fig. 2 D). Taken together, these observations suggest that sfSGK is activated via phosphorylation of the A-loop by PDK1 and of the HM by TORC2, and that the latter is prerequisite for the former. In mammalian cells, PDK1 and TORC2 phosphorylate SGK3 downstream of PI3K (Kobayashi et al., 1999; Tessier and Woodgett, 2006a; Lien et al., 2017; Malik et al., 2018). Consistent with this, the pan-PI3K inhibitor wortmannin blocked both A-loop phosphorylation and the mobility shift of sfSGK (Fig. 2 D), suggesting that this cascade is conserved in the 1-MA–signaling pathway.

To separate fully activated sfSGK from sfSGK phosphorylated only on the HM, we performed Phos-tag SDS-PAGE, in which phosphorylated proteins migrate more slowly than in normal SDS-PAGE (Kinoshita et al., 2006). sfSGK migrated as a single band in unstimulated oocytes (Fig. 2 E, bottom band). Two additional slower-migrating bands were detected after 1-MA stimulation (Fig. 2 E, top and middle bands). The A-loop phosphorylation was detected only in the top band (Fig. 2 E), suggesting that this band corresponds to sfSGK phosphorylated by both PDK1 and TORC2. Indeed, the top band was eliminated by the PDK1 inhibitor BX795 and the TORC1/2 inhibitor pp242, but not by the TORC1 inhibitor rapamycin (Fig. 2 F). In addition, the intensity of the middle band was reduced by pp242, but not by BX795 or rapamycin (Fig. 2 F), suggesting that it represents sfSGK phosphorylated only on the HM by TORC2. Wortmannin abolished all of the mobility shifts (Fig. 2 F), indicating that all phosphorylation detected using Phos-tag depends on PI3K. Taken together, these observations suggest that PDK1 phosphorylates the A-loop of the subset of sfSGK that was prephosphorylated on the HM.

### Activation of sfSGK is required for rapid pH_i increase after 1-MA stimulus

Next, we investigated whether sfSGK is involved in the pH_i increase after 1-MA stimulus. Given that the anti-sfSGK-HM antibody immunoprecipitated sfSGK protein (Fig. 1 C), the binding of the antibody to the HM may be strong enough to sterically block phosphorylation of the HM by TORC2. Indeed, the mobility shift of sfSGK after 1-MA stimulation was blocked in oocytes injected with the antibody (Fig. 3 A). More importantly, the A-loop phosphorylation was also blocked (Fig. 3 A), as observed upon pp242 treatment (Fig. 2 D). In these oocytes, the HM of Akt was phosphorylated as normal (Fig. 3 A). Thus, we concluded that this antibody specifically blocked sfSGK activation; accordingly, hereafter we refer to this antibody as an sfSGK-neutralizing antibody.

To determine whether sfSGK activation is required for the pH_i increase after 1-MA treatment, we injected a pH-sensitive fluorescent dye, 2′,7′-bis-(2-carboxyethyl)-5(and-6)-carboxyfluorescein (BCECF)–dextran, into unstimulated oocytes along with the sfSGK-neutralizing antibody and monitored the pH_i dynamics after 1-MA stimulation via time-lapse recording of the fluorescence ratio. The sfSGK-neutralizing antibody blocked the pH_i increase upon 1-MA stimulation, whereas control IgG did not (Fig. 3 B), indicating that the pH_i increase depends on sfSGK activation.

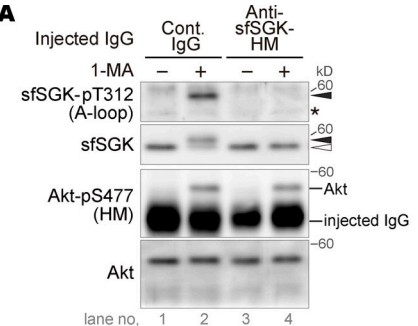

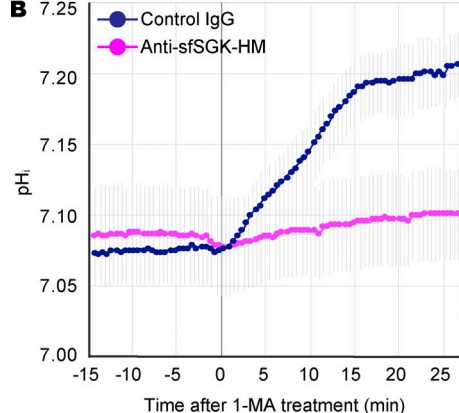

Figure 3. **Activation of sfSGK is required for rapid pH_i increase after 1-MA stimulus. (A)** Unstimulated oocytes were injected with an anti-sfSGK-HM antibody or control IgG, treated with 1-MA for 4 min, and then subjected to immunoblotting. Asterisk, nonspecific bands. Closed and open arrowheads indicate positions of the upper and lower band of sfSGK, respectively. The result shown is representative of three independent experiments. **(B)** BCECF-dextran and either anti-sfSGK-HM antibody or control IgG were coinjected into unstimulated oocytes. After a 1-h incubation, 1-MA was added, and the fluorescence intensity ratio was measured before and after 1-MA addition. pH_i was calculated from the fluorescence intensity ratio and plotted (means ± standard error [SE] of three independent experiments).

### Activation of sfSGK, but not pH_i increase, is required for cyclin B–Cdk1 activation

Previously, we showed that GVBD occurs in isolated oocytes even when the pH_i increase is blocked (Harada et al., 2003). Nonetheless, to our surprise, GVBD was blocked by injection of the sfSGK-neutralizing antibody (Fig. 4, A and B). Consistently, hyperphosphorylation of Cdc25 and dephosphorylation of Tyr15 of Cdk1 were inhibited (Fig. 4 C), indicating that this antibody blocked the signal transduction leading to cyclin B–Cdk1 activation.

To verify that these inhibitory effects were caused by specific inhibition of sfSGK activation, we performed a rescue experiment. Specifically, we replaced Thr479 of sfSGK with Glu (T479E) to mimic a phosphorylated form of the HM, thereby allowing PDK1-dependent A-loop phosphorylation even in the presence of the neutralizing antibody. We coinjected mRNA encoding the mutant sfSGK and the sfSGK-neutralizing antibody into unstimulated oocytes. After a 22-h incubation, the expressed mutant protein was partially phosphorylated on its A-loop even in the absence of 1-MA (Fig. 4 D). This partial activation did not induce GVBD. 1-MA stimulation of

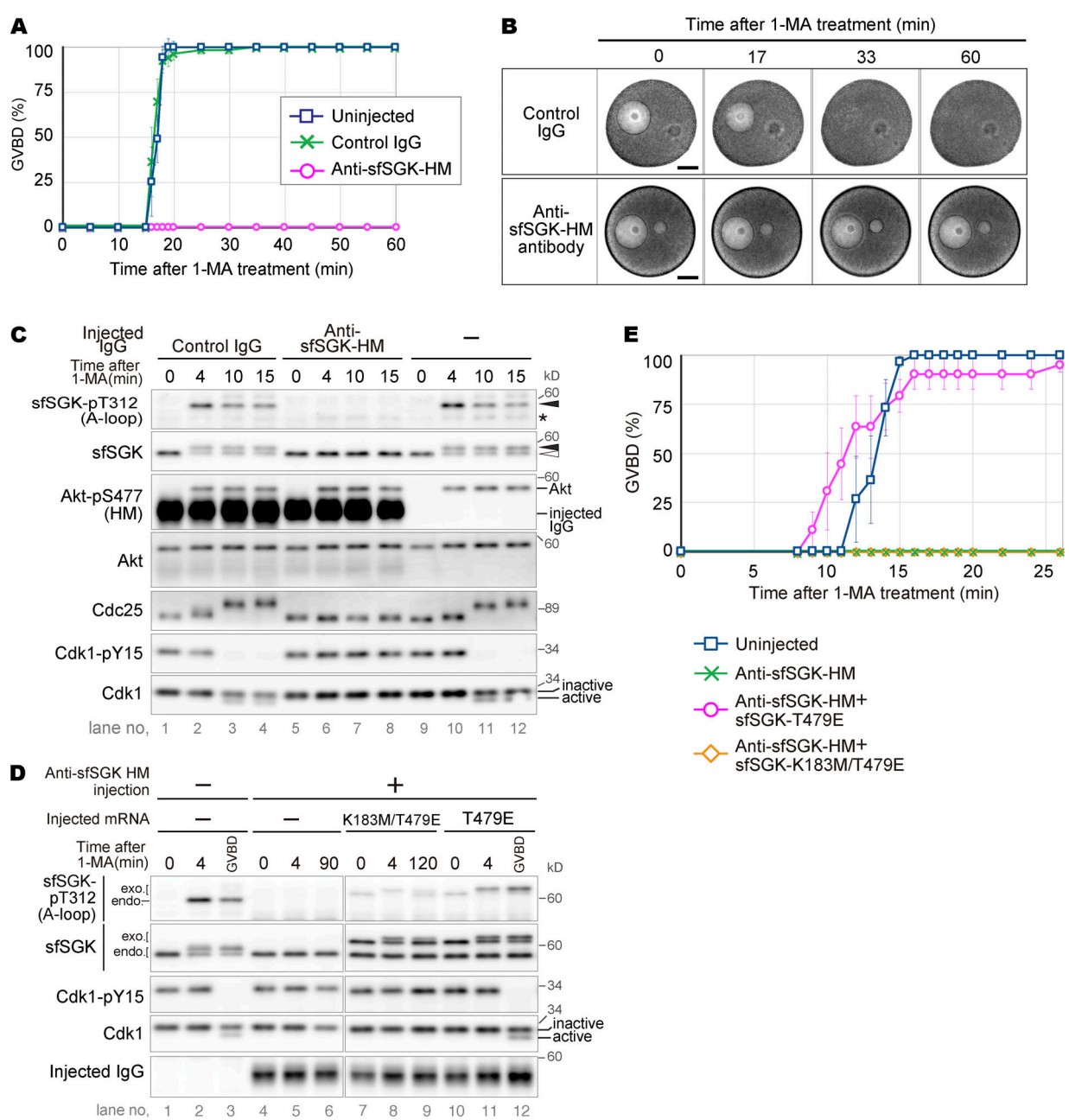

Figure 4. **Activation of sfSGK is required for cyclin B–Cdk1 activation. (A–C)** Unstimulated oocytes were injected with anti-sfSGK-HM antibody or control IgG and treated with 1-MA, followed by monitoring GVBD (A; means ± SE of three independent experiments); imaging by DIC microscopy (B); and immunoblotting (C) at the indicated times. An oil drop introduced along with the antibodies as a mark of the injection can be seen on the right of the GV in B. Bar in B, 50 μm. Asterisk in C, nonspecific bands. Closed and open arrowheads in C indicate positions of the upper and lower bands of sfSGK, respectively. **(D and E)** Unstimulated oocytes were injected with anti-sfSGK-HM antibody, incubated for 1 h, and further injected with mRNA encoding a mutant sfSGK (T479E or K183M/T479E), followed by additional incubation for 22 h. These oocytes were treated with 1-MA, followed by immunoblotting (D) and monitoring GVBD (E; means ± SE of three independent experiments). exo. and endo. in D indicate exogenous and endogenous sfSGK, respectively. Lanes 3 and 12 in D represent oocytes that were collected at the time of GVBD under each condition (∼16 and ∼14 min, respectively). The results shown in B–D are representative of three independent experiments.

these oocytes enhanced A-loop phosphorylation in the T479E mutant, whereas activation of endogenous sfSGK was blocked (Fig. 4 D). Subsequently, activation of cyclin B–Cdk1 (Fig. 4 D) and GVBD (Fig. 4 E) occurred in the T479E mutant-expressing oocytes, whereas it did not when a catalytically inactive version of the mutant (K183M/T479E) was used, indicating that the neutralizing antibody-injected oocytes

were rescued by the activity of the T479E mutant. Thus, we concluded that the inhibitory effect of the antibody is highly specific and that activation of sfSGK is required for cyclin B–Cdk1 activation.

Interestingly, we found that the band of the T479E mutant phosphorylated on the A-loop underwent an upward mobility shift after 1-MA stimulation, indicating that the mutant was

phosphorylated on a site outside the A-loop and the HM. The function of this phosphorylation remains unclear.

Although the $pH_i$ increase and cyclin B–Cdk1 activation were simultaneously blocked by the sfSGK-neutralizing antibody, they were independent of each other. We base this claim on two lines of evidence: (1) the increase in $pH_i$ began before cyclin B–Cdk1 activation (Fig. 2 A and Fig. 3 B) and was not affected by the Cdk1 inhibitor roscovitine (Fig. S1 A), indicating that the $pH_i$ increase is independent of cyclin B–Cdk1 activation; and (2) oocytes underwent GVBD even when the $pH_i$ increase was blocked by an NHE inhibitor or by treatment with sodium-free ASW (Harada et al., 2003), indicating that the cyclin B–Cdk1 activation leading to GVBD is independent of the $pH_i$ increase (see also Fig. 5 A below). Thus, the $pH_i$ increase and cyclin B–Cdk1 activation are mutually independent events.

### Reduced $pH_i$ delays GVBD and blocks invasion of cytoplasmic granules

To elucidate the role of the $pH_i$ increase, we investigated effect of altered $pH_i$ on 1-MA–induced meiotic resumption. To clamp the $pH_i$, oocytes were incubated in sodium-free ASW containing $CH_3COONH_4$ (modified ASW) in which the $pH_i$ was adjusted to the desired value with $CH_3COONH_4$ but did not increase after 1-MA stimulation because of the absence of sodium ion (Moriwaki et al., 2013; Fig. S1 B; see also Materials and methods). We clamped the $pH_i$ at ∼6.7, corresponding to the value in unstimulated ovarian oocytes (Moriwaki et al., 2013); as well as at ∼7.0 and ∼7.2, similar to the values in isolated oocytes before and after 1-MA stimulation, respectively (Moriwaki et al., 2013). First, we examined the effects of $pH_i$ clamping on sfSGK and cyclin B–Cdk1 activation after 1-MA stimulation. The phosphorylation status of sfSGK was basically the same as that in oocytes in ASW. Interestingly, Cdc25 hyperphosphorylation and Cdk1 dephosphorylation (pY15) at lower $pH_i$ occurred a few minutes earlier than those at higher $pH_i$ (Fig. 5 A), suggesting that the signaling leading to cyclin B–Cdk1 activation somewhat prefers lower $pH_i$. This finding further supports that the $pH_i$ increase is not required for cyclin B–Cdk1 activation.

The levels of cyclin B–Cdk1 activity were comparable among oocytes at all of the clamped $pH_i$ values and those in ASW (Fig. S2). Nonetheless, GVBD was delayed for several minutes at a clamped $pH_i$ of 6.7 compared with those at clamped $pH_i$ values of 7.0 and 7.2 (Fig. 5 B), as previously reported (Moriwaki et al., 2013). Under these conditions, the morphology of GVBD was indistinguishable from those in ASW (Fig. S3 and Videos 1, 2, 3, and 4). These results suggest that the GVBD-inducing processes after cyclin B–Cdk1 activation prefer higher $pH_i$.

More importantly, we observed a remarkable defect after GVBD. In ASW and at a clamped $pH_i$ of 7.2, cytoplasmic granules invaded and occupied the entire inner GV area within 15 min and 20–30 min after GVBD, respectively (Fig. 5 C; Videos 1 and 4; and Fig. S4). Thereafter, cyclin B–Cdk1 activity was decreased due to cyclin B degradation at the exit from MI (Fig. S2, A and B). When the $pH_i$ was clamped at 7.0 and 6.7, the amounts of cyclin B protein and cyclin B–Cdk1 activity were maintained over a prolonged period (Fig. S2). This observation is consistent with an inhibitory effect of reduced pH values on cyclin B degradation

in the cell-free oocyte preparation, which we identified as a cause of the MI arrest (Usui et al., 2008). Surprisingly, at a clamped $pH_i$ of 6.7, the cytoplasmic granule invasion proceeded slowly and stalled around 40 min, remaining incomplete even 60 min after GVBD (Fig. 5 C, Video 2, and Fig. S4). At a clamped $pH_i$ of 7.0, invasion proceeded further and finished with a trace of inner GV area within 40–50 min (Fig. 5 C, Video 3, and Fig. S4). These results suggest that the granule invasion is sensitive to $pH_i$; in particular, it is drastically impaired at a $pH_i$ of 6.7.

### Reduced $pH_i$ prevents actin-dependent chromosome transport and microtubule organization for spindle assembly

Next, we focused on dynamic changes in the F-actin and microtubule architecture that occur during the period of cytoplasmic granule invasion. Previous studies demonstrated that just before GVBD, an F-actin shell essential for nuclear envelope (NE) fragmentation forms on the inner surface of the GV (Lénárt et al., 2005; Mori et al., 2014). The shell disappears within 1 min after GVBD, simultaneously with formation of an F-actin meshwork in the entire inner GV area and F-actin patches surrounding the each condensed chromosome (Lénárt et al., 2005). At this time, the chromosomes are randomly scattered in the inner GV area; thereafter, they are transported via contractile flow of the actin meshwork toward the animal pole, where two centrosomes exist near the plasma membrane (Lénárt et al., 2005; Mori et al., 2011; Bun et al., 2018). Subsequently, the actin patches disappear and the transported chromosomes are captured by astral microtubules from the centrosomes, followed by assembly of a spindle with aligned chromosomes (Lénárt et al., 2005; Burdyniuk et al., 2018).

To investigate the effects of $pH_i$ changes on these processes, we fixed $pH_i$-clamped oocytes and visualized the F-actin, microtubules, and chromosomes. We confirmed the presence of condensed chromosomes, two asters, a transient F-actin shell, and an F-actin meshwork in control oocytes in ASW (Fig. 6 A). These structures were also observed in oocytes at all clamped $pH_i$ values (Fig. 6 A). The F-actin patches varied in size (Burdyniuk et al., 2018). In our preparation, although small patches were indistinguishable from the F-actin meshwork, probably due to insufficient resolution, large patches were visible at all clamped $pH_i$ values (Fig. 6 B).

Spindles were assembled at clamped $pH_i$ values of 7.2 and 7.0, albeit with a delay at 7.0 (Fig. 7, A and B). Interestingly, we found unaligned chromosomes in most of the spindles at a clamped $pH_i$ of 7.0, whereas all of the chromosomes were aligned at the midplane at a clamped $pH_i$ of 7.2 (Fig. 7, A and C), suggesting that chromosome alignment is pH sensitive.

When the $pH_i$ was clamped at 6.7, chromosome transport was disrupted. Lénárt et al. (2005) demonstrated that a contractile flow of the F-actin meshwork can gather all of the chromosomes to the animal pole within ∼20 min after GVBD, even upon microtubule disruption by nocodazole. However, although chromosome transport toward the animal pole somewhat progressed over time at a clamped $pH_i$ of 6.7, ungathered chromosomes were still observed even at 60 min after GVBD (Fig. 7, A and D). This finding suggests that actin meshwork–dependent chromosome transport was less efficient.

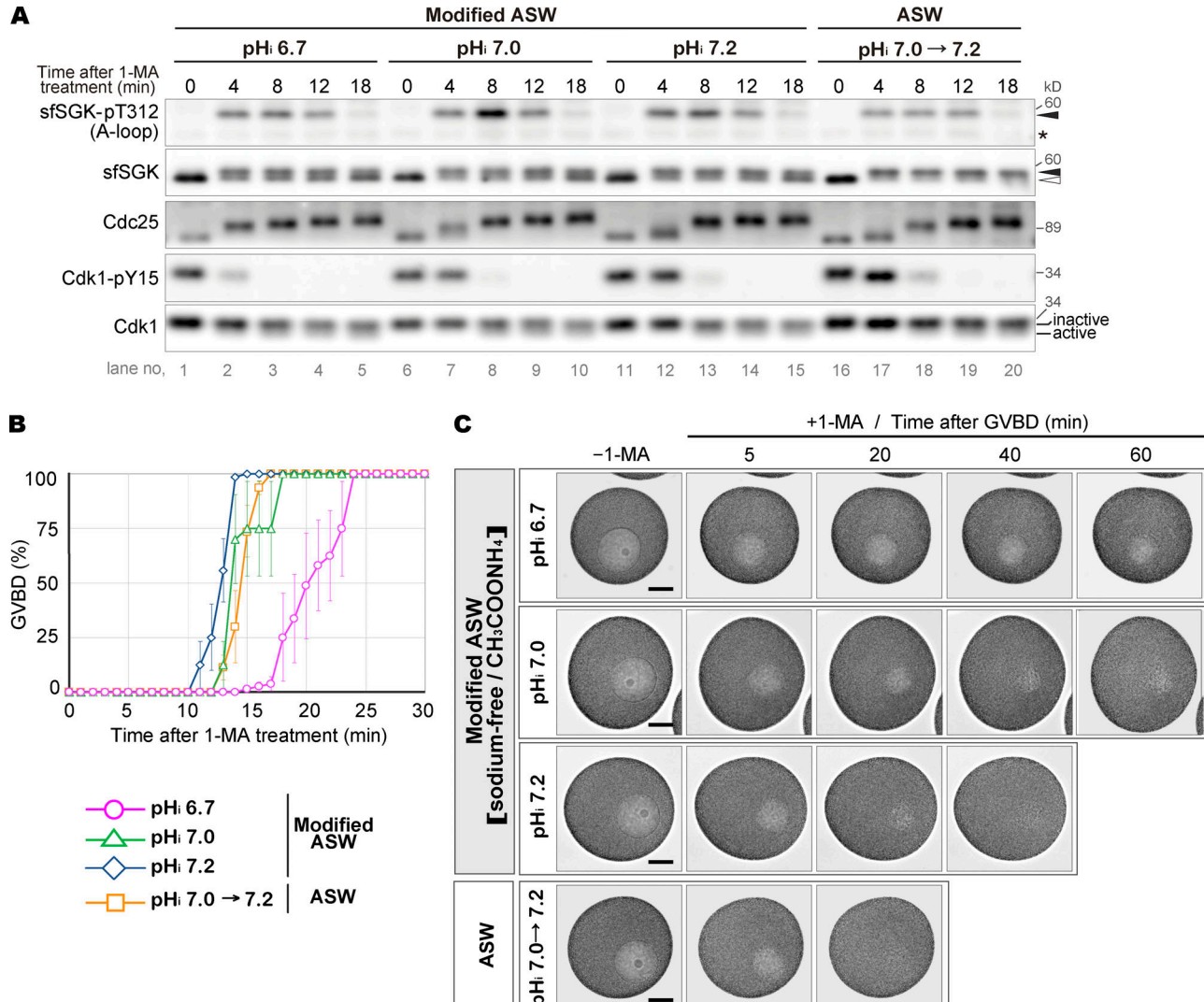

Figure 5. **Reduced pH$_i$ delays GVBD and blocks cytoplasmic granule invasion into the GV region. (A–C)** To clamp the pH$_i$ at ~6.7, 7.0, and 7.2, unstimulated oocytes were incubated with modified ASW for 20 min (see also Fig. S1 B). As a control, unstimulated oocytes were incubated in ASW for 20 min, in which the basal pH$_i$ is ~7.0 but increases to ~7.2 after 1-MA stimulation (Moriwaki et al., 2013; see also Fig. S1 B). These oocytes were stimulated with 1-MA, followed by immunoblotting (A); monitoring GVBD (B; means ± SE of three independent experiments); and time-lapse DIC imaging (C; images captured every 10 s; selected images after GVBD are shown; see Videos 1–4 for complete image sequences). Asterisk in A, nonspecific bands. Closed and open arrowheads in A indicate the positions of the upper and lower sfSGK bands, respectively. Bar in C, 50 µm. GVBD morphology is shown in Fig. S3. Z-stacks acquired after the time-lapse imaging are shown in Fig. S4. The results in A and C are representative of two independent experiments.

The most remarkable defect at a clamped pH$_i$ of 6.7 was found on microtubule organization. Lénárt et al. (2005) also demonstrated that even when actin-dependent transport is inhibited by latrunculin B, the chromosomes close to, but not those far from, the animal pole are captured by astral microtubules originating at the centrosomes and that a spindle is ultimately assembled within a normal time frame, although it lacks the distal chromosomes. However, at a clamped pH$_i$ of 6.7, chromosomes close to the animal pole did not reside between the centrosomes, and a spindle-shaped microtubule structure was not observed even at 60 min after GVBD (Fig. 7, A and B), suggesting that microtubule organization for spindle assembly is perturbed at this pH$_i$.

In light of these data, we concluded that formation of the F-actin structures is pH$_i$-independent but that actin-dependent chromosome transport and microtubule organization are pH$_i$-dependent; in particular, oocytes fail at chromosome gathering and spindle assembly at a pH$_i$ of 6.7. Given that the pH$_i$ of unstimulated ovarian oocytes is around 6.7 (Moriwaki et al., 2013), the pH$_i$ increase mediated by sfSGK is likely a prerequisite for these processes in ovarian oocytes.

## Discussion

This study showed that sfSGK is activated by TORC2 and PDK1 after 1-MA stimulus and that sfSGK is required for pH$_i$ increase and cyclin B–Cdk1 activation. Furthermore, we demonstrated that processes between 1-MA stimulation and spindle assembly have various sensitivities to pH$_i$ changes. In particular,

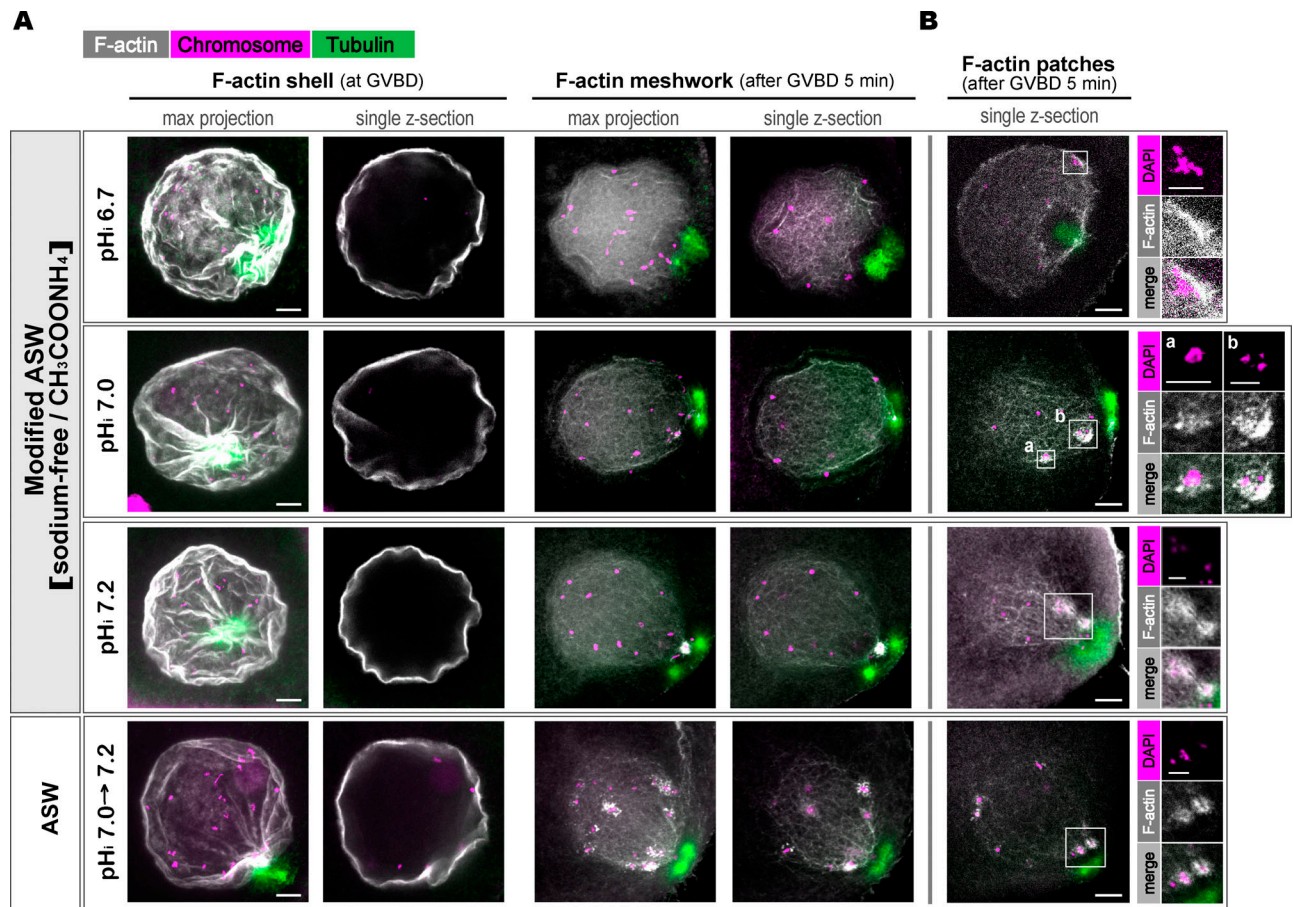

Figure 6. **F-actin shell, meshwork, and patches were formed at all clamped pH$_i$ values. (A and B)** Oocytes stimulated with 1-MA in ASW or in modified ASW to clamp the pH$_i$ at ~7.2, 7.0, or 6.7 were fixed upon GVBD or 5 min after GVBD and then triple-stained with phalloidin for F-actin (gray), anti-α-tubulin antibody for microtubules (green), and DAPI for chromosomes (magenta). Typical images of GV region (max projections and single slices) from two independent experiments are shown in A. Single slices including the actin patches (white boxes with enlarged images) observed 5 min after GVBD are shown in B. Bars in A and B, (main images) 10 µm, (insets) 5 µm.

actin-dependent chromosome transport is less efficient and microtubule organization for spindle assembly is severely disturbed at pH$_i$ 6.7.

On the basis of previous and the present findings, we propose a model for meiotic resumption in ovarian oocytes (Fig. 8). In the body cavity of starfish, ovarian oocytes reside in coelomic fluid, which has relatively high CO$_2$ and low O$_2$ concentrations relative to ASW (Moriwaki et al., 2013). Under these gas conditions, the pH$_i$ of unstimulated oocytes is around 6.7, and it increases after 1-MA stimulation (Moriwaki et al., 2013). In ovarian oocytes, sfSGK is activated after 1-MA stimulation (Fig. 2 C), and it likely induces the pH$_i$ increase and cyclin B–Cdk1 activation (Fig. 8), as demonstrated using isolated oocytes (Figs. 3 and 4). Thereafter, cyclin B–Cdk1 induces GVBD. However, if the pH$_i$ in ovarian oocytes does not increase from 6.7 after 1-MA stimulation, oocytes fail to gather chromosomes and to organize microtubules for spindle assembly, as we observed in isolated oocytes at a clamped pH$_i$ of 6.7 (Figs. 6 and 7). Thus, the sfSGK-dependent pH$_i$ increase would be a prerequisite for processes toward spindle assembly in ovarian oocytes (Fig. 8). We found that the signaling leading to cyclin B–Cdk1 activation prefers lower pH$_i$ (Fig. 5), whereas the processes after cyclin B–Cdk1 activation

prefer higher pH$_i$ (Figs. 5 and 7). Therefore, an sfSGK-dependent transition from a lower pH$_i$ to a higher pH$_i$ in parallel with cyclin B–Cdk1 activation could ensure smooth progression of the molecular events during meiotic resumption in ovarian oocytes.

Although mammalian SGKs are involved in cell proliferation (Hayashi et al., 2001; Gasser et al., 2014), their functions in M-phase remain unclear. Here, we showed that sfSGK is required for cyclin B–Cdk1 activation (Fig. 4). In parallel, Hiraoka et al. demonstrated that sfSGK, rather than Akt, is the major kinase that directly phosphorylates Cdc25 and Myt1 to activate cyclin B–Cdk1 Hiraoka et al. (2019). These are the first reports to demonstrate the involvement of SGK in cyclin B–Cdk1 activation.

Furthermore, we showed that sfSGK is required for the pH$_i$ increase (Fig. 3 B), which depends on sfNHE3 (Harada et al., 2003, 2010; Moriwaki et al., 2013). In cultured human cells, a mechanism of NHE up-regulation is activating phosphorylation of its C terminus (Takahashi et al., 1999; Wang et al., 2005). Although it remains unclear how sfSGK up-regulates sfNHE3 in starfish oocytes, the C terminus of sfNHE3 is phosphorylated by human SGK1 in vitro (Harada et al., 2010). Therefore, we speculate that sfSGK activates sfNHE3 via direct phosphorylation of its C terminus.

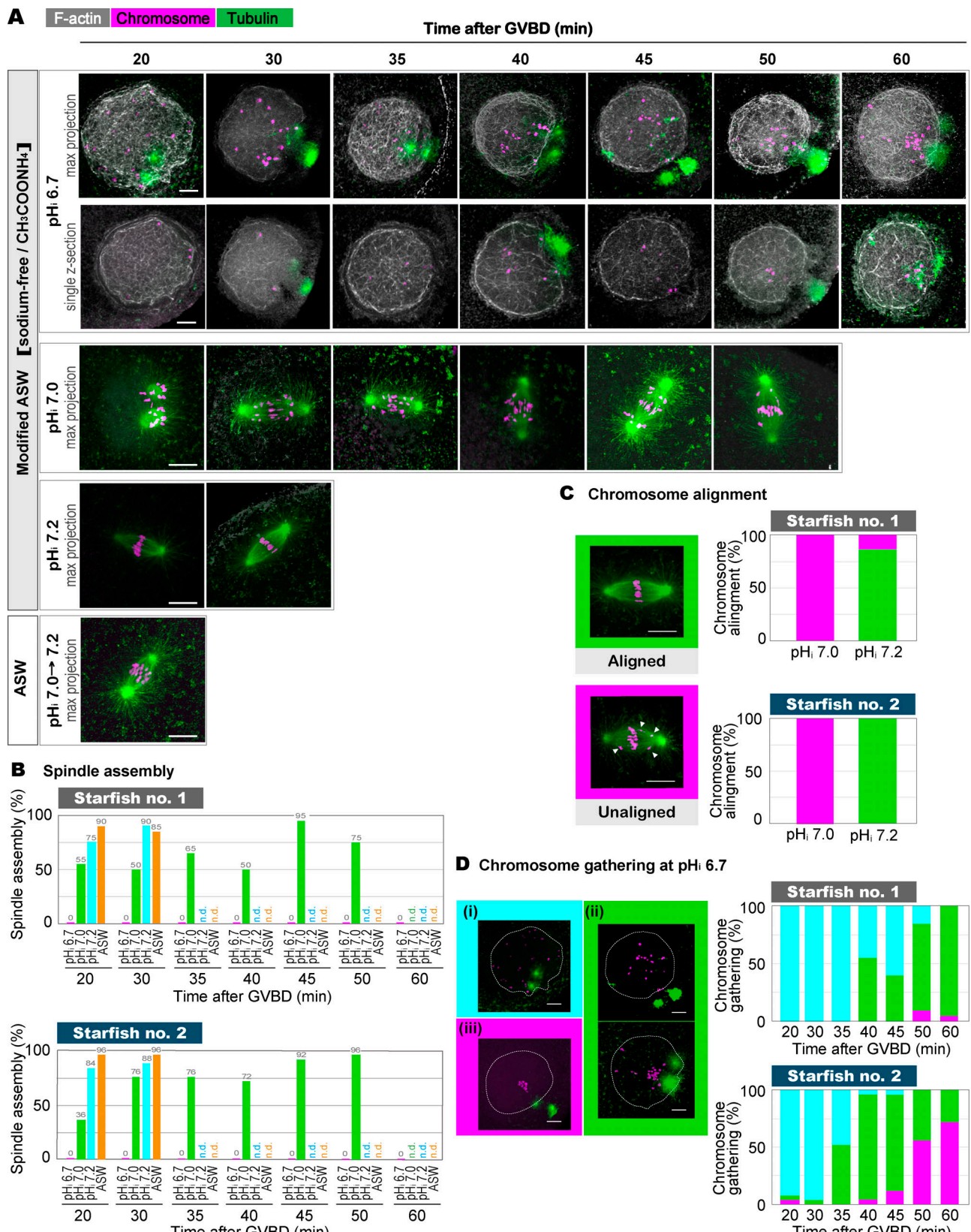

**Figure 7. Reduced pH$_i$ perturbed chromosome transport and microtubule organization for spindle assembly. (A)** Oocytes were treated and stained as in Fig. 6 at the indicated time points after GVBD. Typical images of GV region or spindles (max projection and/or single slices) from two independent experiments are shown. GVBD occurred around 17 min after 1-MA stimulation in ASW and around 13, 10, and 18 min at clamped pH$_i$ values of 7.2, 7.0, and 6.7, respectively. Bars, 10 µm. The stained samples were also used in B–D. **(B)** Oocytes with spindles were counted at the indicated times after GVBD. n.d., not

determined. **(C)** Spindles assembled at clamped $pH_i$ values of 7.0 (45–50 min after GVBD) and 7.2 (20–30 min after GVBD) were checked for chromosome alignment (aligned or unaligned; typical images are shown). Arrowheads indicate unaligned chromosomes. Bars, 10 µm. **(D)** At the indicated time points after GVBD, oocytes at a clamped $pH_i$ of 6.7 were classified according to their chromosome distribution (typical images are shown): (i, cyan) scattered within the entire GV region; (ii, green) somewhat gathered toward the animal pole, but not in a compact group; (iii, magenta) gathered in a compact group near the animal pole. Dotted lines, boundary of the GV region. Bars, 10 µm. In B–D, results from two females (starfish 1 and 2) are shown. In B–D, 20 and 25 oocytes were observed at each time point in starfish 1 and 2, respectively.

We observed that invasion of cytoplasmic granules into the GV region after GVBD stalled at a clamped $pH_i$ of 6.7 (Fig. 5). One possible explanation is that incomplete NE fragmentation occurred, and disrupted granule invasion. Efficient NE fragmentation requires small F-actin protrusions formed on the F-actin shell (Mori et al., 2014; Wesolowska et al., 2018 *Preprint*). Therefore, formation and/or function of these protrusions may be perturbed at a $pH_i$ of 6.7. At the resolution of our fluorescent images, the shell was visible, but the protrusions were not (Fig. 6). Thus, imaging with higher resolution and analysis at ultrastructural levels will be required to test this prediction.

The F-actin meshwork formed at a clamped $pH_i$ of 6.7, but subsequent chromosome gathering required more time and was incomplete (Fig. 7). Meshwork-dependent chromosome transport relies on an actin disassembly-driven contractile flow of the meshwork (Bun et al., 2018). Thus, proteins that control actin disassembly might be involved in chromosome gathering and could be sensitive to $pH_i$ changes. Actin depolymerizing factor/cofilin family proteins promote actin filament disassembly in mammalian cultured cells (Hotulainen et al., 2005; Kanellos and Frame, 2016) and show higher activity at higher pH values (Yeoh et al., 2002). The possible roles of these proteins in a $pH_i$-sensing mechanism in starfish oocytes would be an intriguing topic for future studies.

Microtubule organization was severely disturbed at a low $pH_i$ (Fig. 7). We speculate that this defect is a consequence of direct effects of pH on microtubule dynamics because pH affects microtubule polymerization/depolymerization in vitro (Regula et al., 1981). pH can also affect microtubule organization in vivo in green algae (Liu et al., 2017) and sea urchin eggs (Harris and Clason, 1992). Furthermore, the mitotic spindles were poorly organized at $pH_i$ 6.3 in fertilized sand dollar eggs (Watanabe et al., 1997). Thus, defective spindle assembly at reduced $pH_i$ values seems to be a common feature in meiosis and mitosis.

The $pH_i$ in ovarian oocytes after 1-MA stimulation was estimated to be ~6.9 (Moriwaki et al., 2013). The ovarian oocytes undergo the MI arrest after GVBD (Harada et al., 2003; Moriwaki et al., 2013). Changes in the environmental $CO_2$ and $O_2$ concentrations at spawning induce a further increase in the $pH_i$ to ~7.2, leading to release from the arrest (Moriwaki et al., 2013). In this scenario, we previously claimed that the stage of the MI arrest in ovarian oocytes is mainly metaphase because the majority of oocytes (~70%) soon after spawning were at metaphase (Harada et al., 2003). In this observation by Harada et al. (2003), the remaining ~30% of the oocytes soon after spawning were still at prometaphase. Considering the presence of unaligned chromosomes in the spindles at a clamped $pH_i$ of ~7.0 (Fig. 7), more ovarian oocytes may stay at late prometaphase than at metaphase; "MI-arrested starfish oocytes" seems to include oocytes at late prometaphase and metaphase.

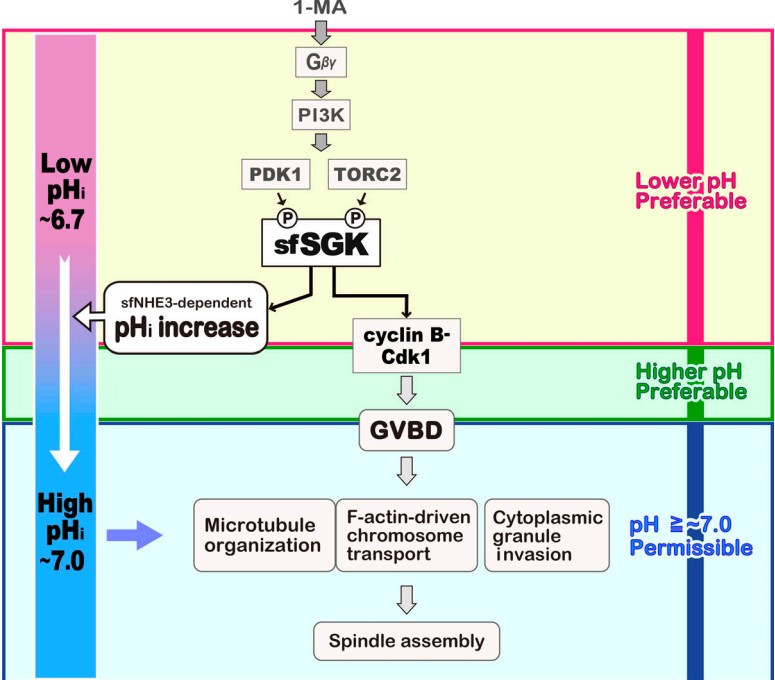

Figure 8. **Model for sfSGK-dependent meiotic resumption in ovarian oocytes.** A model for 1-MA–induced meiotic resumption in ovarian oocytes was prepared on the basis of previous and current findings. 1-MA stimulus induces Gβγ-dependent PI3K activation. In a manner that depends on PI3K, PDK1 and TORC2 activate sfSGK. Activation of sfSGK is essential for cyclin B–Cdk1 activation and sfNHE3-dependent $pH_i$ increase. These SGK-dependent pathways prefer lower $pH_i$ (magenta box). Processes leading to GVBD after cyclin B–Cdk1 activation prefer higher $pH_i$ (green box). After GVBD, actin-dependent chromosome gathering and microtubule organization for spindle assembly as well as cytoplasmic granule invasion into the GV region require a $pH_i$ of ~7.0 or higher (cyan box) although spindles contains unaligned chromosomes at $pH_i$ ~7.0. Because the $pH_i$ of ovarian oocytes before 1-MA stimulation in the body cavity is ~6.7, the sfSGK-dependent $pH_i$ increase accelerates GVBD and is required for the processes leading to meiotic spindle assembly in ovarian oocytes. By contrast, the $pH_i$ of isolated oocytes before 1-MA stimulation in ASW is ~7.0, which is already permissive for the processes toward spindle assembly.

Although it remains to be determined which is the actual main stage of the MI arrest, an important feature could be that oocytes wait for spawning at stages before onset of anaphase I. Insemination after the first polar body formation tends to result in polyspermy (Miyazaki and Hirai, 1979; Usui et al., 2008). Therefore, fertilization before polar body formation is important for monospermy. To ensure this timing of fertilization, the MI (prometaphase/metaphase) arrest must be maintained until spawning (Usui et al., 2008). To this end, the sfSGK-dependent $pH_i$ increase should not exceed 7.0, because release from the arrest occurs at $pH_i > 7.0$ (Harada et al., 2003; Oita et al., 2004; Usui et al., 2008; Moriwaki et al., 2013). A-loop phosphorylation of sfSGK became undetectable around the time of the end of $pH_i$ increase (Fig. 2, A and C; and Fig. 3 B). Therefore, sfSGK inactivation may be important to stop the $pH_i$ increase to maintain the arrest. Interestingly, the A-loop was dephosphorylated even in the presence of phosphorylated HM (Fig. 2, A and C), implying the presence of regulatory mechanism of A-loop dephosphorylation that is not associated with the HM dephosphorylation (for example, up-regulation of an A-loop phosphatase). To our knowledge, regulation of SGK inactivation has never been investigated in any animal, and this issue should be examined in the future. Furthermore, our observation of unaligned chromosomes at a clamped $pH_i$ of ~7.0 implies that this $pH_i$ value might be a somewhat inadequate condition for the establishment and/or maintenance of bi-oriented attachment. If this is the case, anaphase onset before spawning might increase probability of aneuploidy, and arrest at stages before anaphase onset would protect oocytes from this potential risk of aneuploidy. The $pH_i$ increase at spawning would allow chromosome alignment before anaphase onset.

Beyond starfish, $pH_i$ increases in oocytes after release from ProI arrest in frog (Lee and Steinhardt, 1981), urodele (Rodeau and Vilain, 1987), and surf clam (Dubé and Eckberg, 1997). In *Xenopus laevis* oocytes, a $pH_i$ increase after hormonal stimulation (Lee and Steinhardt, 1981; Stith and Maller, 1985; Rezai et al., 1994) has been suggested to accelerate cyclin B–Cdk1 activation by promoting accumulation of Mos protein, which participates in cyclin B–Cdk1 activation (Sellier et al., 2006). By contrast, we found that cyclin B–Cdk1 activation was somewhat accelerated at a reduced $pH_i$ in starfish oocytes. This inconsistency might be due to differences in the mechanisms inducing cyclin B–Cdk1 activation. In starfish oocytes, cyclin B–Cdk1 activation does not require new protein synthesis (Tachibana et al., 1997), including Mos (Tachibana et al., 2000). A $pH_i$ increase in *Xenopus* oocytes also plays a role in GV migration to the animal pole before GVBD (Flament et al., 1996). Such migration does not occur in starfish oocytes because the GV already resides at the animal pole even in ProI-arrested oocytes. Instead, we found that chromosome transport and microtubule organization depend on $pH_i$. Thus, starfish and *Xenopus* oocytes provide suitable models for understanding different aspects of $pH_i$-dependent regulation of the meiotic cell cycles.

In summary, we showed that sfSGK is required for $pH_i$ increase and cyclin B–Cdk1 activation in starfish oocytes, and that the processes toward spindle assembly after GVBD are $pH_i$ sensitive. These findings reveal a novel role for SGK in controlling meiotic cell cycles and emphasize the importance of tight $pH_i$ regulation in oocyte meiosis.

## Materials and methods

### Oocyte preparation

Starfish (*Asterina pectinifera*) were collected on the Pacific coast of Japan and kept in laboratory aquariums supplied with circulating seawater at 14°C. Fully grown ProI-arrested oocytes were isolated from ovaries and treated with calcium-free ASW (476 mM NaCl, 10 mM KCl, 36 mM $MgCl_2$, 18 mM $MgSO_4$, and 20 mM $H_3BO_3$, pH 8.2) to remove follicle cells. Subsequently, the oocytes were kept in ASW (462 mM NaCl, 10 mM $CaCl_2$, 10 mM KCl, 36 mM $MgCl_2$, 18 mM $MgSO_4$, and 20 mM $H_3BO_3$, pH 8.2) until use. For 1-MA stimulation, the isolated oocytes were incubated with ASW containing 0.5 µM 1-MA at 23°C, and then collected for analysis. To induce meiotic resumption of ovarian oocytes in the body cavity, 1 ml of calcium-free ASW without boric acid (462 mM NaCl, 10 mM KCl, 36 mM $MgCl_2$, and 17 mM $MgSO_4$) containing 0.5 mM 1-MA was injected into the body cavity of female starfish. Pieces of ovary containing stimulated oocytes were directly recovered from the body cavity, and then immediately placed into a Laemmli sample buffer (LSB: 0.0625 M Tris-HCl, pH 6.8, 10% glycerol, 2% SDS, 5% 2-mercaptoethanol, and 0.0025% bromophenol blue). To compare proteins in germinal epithelium with those in oocytes, a piece of ovary was isolated from the body cavity and separated into germinal epithelium and oocytes. These fractions were separately recovered into different tubes, each of which contained an equal volume of LSB, boiled at 95°C for 5 min, and subjected to immunoblotting analysis. Unless otherwise indicated, isolated oocytes in ASW were used. Use of the modified ASW or ovarian oocytes is indicated in the main text and/or legends.

### cDNA cloning of sfSGK

To isolate a cDNA encoding the starfish homologue of SGK, total mRNA was isolated from ProI-arrested ovarian oocytes using RNAwiz (Ambion, Thermo Fisher Scientific). The first-strand cDNA library was prepared from total mRNA using the TAKARA RNA PCR Kit (AMV) Ver.2.1 and Prime Script Reverse transcription (Takara Bio) with random 9-mers. A 250-bp cDNA fragment (fragment 1) was first obtained by RT-PCR with degenerate primers designed based on amino acid sequences conserved among human, mouse, sea urchin, and *Caenorhabditis elegans* SGKs (forward: YAVKVL, 5′-TAYGCNGTNAARGTNYT-3′, and reverse: FYAAEIA, 5′-AARATRCGNCGNCTYTCDCG-3′). Because 3′ and 5′ rapid amplification of cDNA ends (RACE) with specific primers designed against fragment 1 were unsuccessful, we obtained another fragment (fragment 2) that partially overlapped with fragment 1, as follows: a forward specific primer (5′-CCCGACAAGCTCTAC-3′) was designed against fragment 1, and a reverse degenerate primer (GLPPFY, 5′-CCNRANGGNGGNA ARAT-3′) was designed against a conserved amino acid sequence in the SGK alignment that is located in a region ~85 aa C-terminal to the region of the first reverse degenerate primer. The 360-bp fragment 2 was obtained by RT-PCR using these

primers. Next, to identify the 3′ end, another first-strand cDNA library was generated from the total mRNA using RNA PCR Kit (AMV) v.2.1 and Prime Script Reverse transcription (both Takara Bio) using the oligo dT-Adaptor primer from the kit. Then, RT-PCR was performed on the cDNA library using a specific forward primer designed against the 5′ region of fragment 2 (5′-CCCCAGAAGTCTTGAAGAA-3′) and the adaptor primer (5′-GTTTTCCCAGTCACGAC-3′). Using the RT-PCR product as a template, nested PCR was performed with another forward specific primer (5′-GGAGTATGATCGTAGTGTAG-3′) and the adaptor primer, resulting in amplification of an ~600 bp 3′ RACE product. Next, the 5′ ends were identified by RACE using 5′-Full RACE Core Set (Takara Bio) with specific primers (reverse for generating first-strand cDNA: 5′-CCAAAGTCTGTC AAC-3′; forward for first PCR: 5′-CAGAGATTTGAAGCCGGA G-3′, reverse for first PCR: 5′-GTTCAGGGAAGCATCTCTTCT-3′; forward for nested PCR: 5′-GCCGGAGAACATTTTGATTG-3′, reverse for nested PCR: 5′CTCTCTCTGCAGATGGAAGA-3′), resulting in amplification of a 1.5-kb 5′ RACE product. The full-length ORF was PCR amplified from the cDNA library generated using random 9-mers with pfx50 DNA polymerase (Thermo Fisher Scientific) and specific primers (forward: 5′-GGGGCCCCT GGGATCCATGGCTGCCAACAGTACTAG-3′; reverse: 5′-GTCGACCC GGGAATTCTTAGTCGTCAAATGCCTCG-3′). The amplified cDNA was inserted into vector pGEX-6P-3 (GE Healthcare) using the In-Fusion HD Cloning Kit (Takara Bio) and sequenced. The DNA sequence of the obtained sfSGK-encoding cDNA was submitted to the DNA Data Bank of Japan/European Molecular Biology Laboratory/GenBank databases under accession no. LC430700. An alignment of the predicted amino acid sequence of sfSGK with that of human SGK3 (NCBI accession no. NP_001028750.1) was made using the ALAdeGAP alignment software (Hijikata et al., 2011).

### Immunoprecipitation
For preparation of oocyte extracts, oocytes were recovered in ASW before and after 1-MA treatment. After freezing in liquid N$_2$, lysis buffer (80 mM β-glycerophosphate, 20 mM EGTA, 10 mM MOPS, pH 7.0, 100 mM sucrose, 100 mM KCl, 1 mM DTT, 0.5% NP-40, 1% protease inhibitor cocktail [Nacalai Tesque], and 1% phosphatase inhibitor cocktails 1 and 2 [Sigma-Aldrich]) was added to the oocytes, followed by vortexing and centrifugation at 12,500 × $g$ for 15 min. The supernatant was used as the extract. For immunoprecipitation, 15 µl of oocyte extract (1 oocyte per 1 µl) was added to 3 µl of protein A–Sepharose 4B (Sigma-Aldrich) to which 7.0 µg of anti-sfSGK-HM antibody or control IgG had been preadsorbed and cross-linked using borate buffer containing disuccinimidyl suberate (25 mM disuccinimidyl suberate, 0.2 M H$_3$BO$_3$, pH 9.0, and 0.2 M NaCl), followed by incubation for 90 min on ice. Beads and flow-through were separated, and LSB was added. The samples were boiled at 95°C for 5 min and subjected to immunoblotting analysis.

### Chemicals
Wortmannin (LC Laboratories), pp242 (Cayman Chemical), BX795 (Enzo Life Sciences), and roscovitine (Merck) were dissolved in DMSO at 20 mM as a stock solution, and used at final concentrations of 40, 2, 6, and 30 µM, respectively. Rapamycin (FUJIFILM Wako Pure Chemical) was dissolved in DMSO at 20 mM as a stock solution. It was used at a final concentration of 20 µM, a concentration that was previously shown to be sufficient for TORC1 inhibition in starfish oocytes (Hiraoka et al., 2011).

### DNA constructs
A mutant sfSGK in which Thr479 was replaced with Glu was generated by PCR using PrimeSTAR MAX Polymerase (Takara Bio) with primers containing the point mutation (forward: 5′-GGCTTCGAATATGTCCCACCAAGCGAG-3′, and reverse: 5′-GAC ATATTCGAAGCCAGTAAAGGCAGAG-3′). To generate the catalytically inactive mutant, Lys183, a conserved residue important for ATP binding, was replaced with Met using primers containing the point mutation (forward: 5′-GCTGTCATGGTCTTG GAAAAGAAAG-3′, and reverse: 5′-CAAGACCATGACAGCGTA CAC-3′). The mutated cDNAs were inserted with the *A. pectinifera* cyclin B Kozak sequence (5′-TACAAT-3′) and a C-terminal 3× FLAG tag into the pSP64-S–based vector in which the SP6 promoter was replaced with a T7 promoter (Hiraoka et al., 2016).

### Antibodies
The anti-sfSGK-ΔN50 rabbit polyclonal antibody was raised against a Nus- and His-tagged recombinant sfSGK fragment lacking the N-terminal 50 aa including the PX domain. The antibody was purified from the antiserum using polyvinylidene difluoride membrane on which a recombinant GST-tagged sfSGK protein had been blotted. The anti-sfSGK-HM rabbit polyclonal antibody was raised against a synthetic peptide representing the 17 C-terminal amino acids containing the HM (Ser473-Asp489) of sfSGK and affinity-purified using antigen peptide–conjugated Sulfo Link Coupling Resin (Thermo Fisher Scientific). The control IgG was purified from rabbit preimmune serum using protein A–Sepharose 4B (Sigma-Aldrich). For microinjection, the anti-sfSGK-HM antibody and control IgG were further concentrated using a 50-kD cut-off Amicon Ultra (Merck), and the buffer was replaced with injection buffer (0.05% NP-40 in PBS, pH 7.2). Preparation of anti-starfish Akt (sfAkt) phospho-Ser477 affinity-purified rabbit polyclonal antibody (Hiraoka et al., 2011; antigen, QFEKFpSYSGDK), anti-sfAkt-C-terminal fragment affinity-purified rabbit polyclonal antibody (Okumura et al., 2002; antigen, C-terminal 88 aa fragment of sfAkt), anti-starfish Cdc25 (sfCdc25) rabbit polyclonal antiserum (Okumura et al., 1996; antigen, C-terminal 153 aa fragment), anti-starfish cyclin B affinity-purified rabbit polyclonal antibody (Okano-Uchida et al., 1998; antigen, full-length starfish cyclin B), and anti-PSTAIR antibody for Cdk1 detection (Hiraoka et al., 2011) were described previously. The anti-Cdk1 phospho-Tyr15 antibody and anti-human phospho-SGK rabbit polyclonal antibody (p-SGK [Thr 256]-R) were purchased from Cell Signaling Technology (9111S) and Santa Cruz Biotechnology (sc-16744-R), respectively. Peroxidase-conjugated anti-rabbit IgG and anti-mouse IgG, used as secondary antibodies in immunoblotting, were purchased from GE Healthcare and Dako, respectively.

### Microinjection
Microinjection was performed as previously described (Kishimoto, 1986; Chiba et al., 1992). The anti-sfSGK-HM

antibody or control IgG was injected into unstimulated oocytes at 23°C at a final concentration in the oocytes of 65.2 µg/ml, followed by incubation for 1 h. For expression of the mutant sfSGK constructs (T479E and K183M/T479E), mRNAs encoding each mutant were transcribed from the pSP64-S–based vector construct and subsequently polyadenylated at their 3′ termini using the mMESSAGE mMACHINE kit and the Poly(A) Tailing Kit (Thermo Fisher Scientific). They were then dissolved in water and injected into unstimulated oocytes (15 pg per oocyte). After further incubation for 22 h at 20°C, these oocytes were treated with 0.5 µM 1-MA and recovered for immunoblotting analysis. The rest of the oocytes were observed to count GVBD.

## Immunoblotting

Oocytes (five for normal SDS-PAGE and six for Phos-tag-SDS-PAGE) were recovered in 3 µl of seawater, added directly to 3 µl of 2× LSB, and immediately frozen in liquid $N_2$. After thawing and boiling for 5 min at 95°C, proteins were separated on polyacrylamide gels (8% for sfSGK only and 8.5% for sfSGK and Cdk1). For Phos-tag SDS-PAGE, 8% polyacrylamide gel containing 60 µM Phos-tag acrylamide (FUJIFILM Wako Pure Chemical) and 600 µM $ZnCl_2$ was used. Immunoblotting was performed as previously described (Hiraoka et al., 2016). Antibodies used for immunoblotting were as follows: anti-sfSGK-ΔN50 (1:2,000, in TBS containing 0.1% Tween 20 [TBS-T]), anti-sfSGK-HM (1:1,000, in TBS-T), anti-human phospho-SGK (1:400, in TBS-T), anti-sfAkt phospho-Ser477 (1:600, in Can Get Signal Immunoreaction Enhancer Solution [Can Get sol.] 1, TOYOBO), anti-sfAkt C-terminal fragment (1:400, in Can Get sol. 1), anti-sfCdc25 (1:2,000, in Can Get sol. 1), anti-starfish cyclin B (1:1,000, in Can Get sol. 1), anti-Cdk1 phospho-Tyr15 (1:1,000, in Can Get sol. 1), and anti-PSTAIR (1:50,000, in Can Get sol. 1). HRP-conjugated anti-rabbit IgG (1:2,000, in TBS-T or Can Get sol. 2) and anti-mouse IgG (1:2,000, in Can Get sol. 2) were used as secondary antibodies. Proteins reacting with the antibodies were visualized with ECL prime (GE Healthcare), and digital images were acquired on a LAS4000 mini imager (FUJIFILM Wako Pure Chemical).

## Measurement of pH$_i$ with BCECF-dextran

A dextran (10-kD) conjugate of BCECF (Thermo Fisher Scientific) was dissolved at a concentration of 2 mM in injection buffer (100 mM potassium aspartate, 10 mM Hepes, and 0.05% NP-40, pH 7.2) and injected into unstimulated oocytes (20 pl per oocyte). After incubation for 1 h, the $pH_i$ of these oocytes was measured under various conditions as described below. The $pH_i$ measurements with BCECF-dextran were made by determining the pH-dependent ratio of its emission intensity (detected at 535 nm) when excited at 495 nm versus its isosbestic point of 436 nm. All fluorescence images were captured and analyzed using an ECLIPSE Ti-U fluorescent microscope (Nikon Instech) equipped with a 4× 0.20 NA CFI super Fluor lens (Nikon Instech), connected to a CMOS camera (ORCA-Flash2.8, C11440-10C; Hamamatsu Photonics), and controlled by an HCImage processing system (Hamamatsu Photonics). Excitation light from a xenon lamp alternated between 436 nm and 495 nm under computational control. The emitted light passed through a dichroic beam splitter at 505 nm and through a 510–560 nm emission filter.

After image acquisition, a region of interest (ROI) surrounding the cytoplasm was defined for each oocyte using HCImage (Hamamatsu Photonics; Fig. S5 A). The system automatically measured the averaged fluorescence intensity inside each ROI and calculated the ratio of the averaged intensities between excitation at 436 nm and 495 nm. The intensity ratios were converted into $pH_i$ values as basically described in our previous study (Moriwaki et al., 2013). A detailed explanation of the conversion follows (see also Fig. S5 for examples of $pH_i$ measurements).

The steady-state $pH_i$ levels of oocytes were determined by injecting pH-standard solutions with various pH values (0.5 M Pipes for pH 6.40–6.75, and 0.5 M Hepes for pH 6.80–7.40; pH was adjusted by KOH) into BCECF-injected oocytes (Fig. S5 B). When the actual $pH_i$ is above or below the known pH of the injected standard solutions, the $pH_i$ falls or rises, respectively. These changes can be detected as a decrease or increase in the fluorescence intensity ratio of BCECF (Fig. S5 B). When the actual $pH_i$ is equal to that of the injected solutions, neither the $pH_i$ (Fig. S5 B) nor the intensity ratio change. In this method, $pH_i$ differences as little as 0.05 are detectable.

For time course measurements, we took advantage of modified ASW containing $CH_3COONH_4$. When unstimulated oocytes were placed in modified ASW, the steady-state $pH_i$ changes depending on the pH of the modified ASW (see Clamping $pH_i$ of oocytes section below). By estimating the steady-state $pH_i$ levels as described above, we previously found that the $pH_i$ in oocytes in modified ASW is ∼0.2 units higher on average than the pH of the modified ASW (Moriwaki et al., 2013). In addition, there is a linear correlation between the intensity ratios and $pH_i$ values at least in a $pH_i$ range of 6.6 to 7.5 (Fig. S5 C). Based on these principles, we could determine the $pH_i$ values under experimental conditions. To determine the $pH_i$, the intensity ratio was measured under three conditions: two standard conditions and an experimental condition. The first standard condition is unstimulated oocytes in pH 6.4–modified ASW, which provides a fluorescence ratio corresponding to a $pH_i$ of ∼6.6 (Fig. S5 D, group 1). The second standard condition is unstimulated oocytes in pH 7.3–modified ASW, which provides a fluorescence ratio corresponding to a $pH_i$ of apprximately 7.5. Plotting these fluorescence ratios against the $pH_i$ gives a standard linear function (Fig. S5 E). Time course measurements of the intensity ratios under experimental conditions were converted to $pH_i$ values using this standard linear function (Fig. S5 F). Under each condition, we normally measured the fluorescence ratio in 8–16 oocytes and used the averaged ratio to derive the standard function and to obtain $pH_i$ values under the experimental condition (Fig. S5, D–F). The experimental conditions were as follows: unstimulated oocytes injected with either sfSGK-HM antibody or control IgG (final 65.2 µg/ml in an oocyte), incubated for 1 h, and then treated with 1-MA; unstimulated oocytes incubated for 1 h in ASW in the presence or absence of roscovitine and then treated with 1-MA; unstimulated oocytes incubated in ASW or modified ASW for 20 min and then treated with 1-MA. The pH of the ASW, the modified ASW, and the standard solutions was adjusted at 23°C using a Navi h pH meter (HORIBA), which has a resolution of 0.001 pH units. All experiments associated with $pH_i$ measurement were performed at 23°C.

## Clamping pH$_i$ of oocytes

NH$_4$Cl dissolved in seawater forms NH$_3$, which crosses the cell membrane easily and binds H$^+$, causing an increase in pH$_i$. In contrast, when CH$_3$COONa is dissolved in seawater, it forms CH$_3$COOH, which crosses cell membrane easily and releases H$^+$ into the cell, causing pH$_i$ to decrease. By applying this principle, Hamaguchi et al. (1997) found that extracellular pH (pH$_o$) and pH$_i$ were almost equal when sea urchin eggs were treated with ASW containing 20 mM CH$_3$COONa and NH$_4$Cl. We showed previously that when starfish oocytes are incubated in sodium-free ASW containing CH$_3$COONH$_4$ at various pH values (modified ASW: 480 mM choline chloride, 55 mM MgCl$_2$, 5 mM KCl, 10 mM Pipes, 10 mM Hepes, 20 mM CH$_3$COONH$_4$, and 9.2 mM CaCl$_2$), the pH$_i$ of oocytes was ~0.2 units higher on average than the pH$_o$ (Moriwaki et al., 2013). Hence, to clamp the pH$_i$ of oocytes at ~6.7, 7.0, and 7.2, unstimulated oocytes were incubated with modified ASW with a pH of 6.5, 6.8, and 7.0, respectively, for 20 min. These oocytes were then treated with 1-MA, followed by observation of GVBD, immunoblotting analysis, H1 kinase assay, or fluorescence imaging. It should be noted that because sodium ion is not present in modified ASW, the sfNHE3-dependent pH$_i$ increase mediated by Na$^+$/H$^+$ exchange does not occur such that the pH$_i$ is maintained at the clamped value even after 1-MA treatment (see Fig. S1 B). As a control, isolated oocytes were incubated in ASW (pH$_o$ 8.2), in which the pH$_i$ of unstimulated oocytes was ~7.0. Because ASW contains sodium ions, a sfNHE3-dependent pH$_i$ increase to 7.2–7.3 occurs after 1-MA treatment (Moriwaki et al., 2013; see also Fig. S1 B). All experiments with pH$_i$ clamping were performed at 23°C.

## Oocyte observation

GVBD was visually judged by focusing the microscope on the equatorial plane of the GV: the rim of intact GV appears as a sharp line in unstimulated oocytes, but becomes fuzzy at GVBD (Lénárt et al., 2003; see also Fig. S3). All DIC images and DIC image sequences were captured on an ECLIPSE Ti-U microscope by using a 20× 0.45 NA Plan Fluor lens (Nikon Instech), which was connected to a CMOS camera (ORCA-Flash2.8, C11440-10C; Hamamatsu Photonics) controlled by a HCImage processing system (Hamamatsu Photonics). For time-lapse recording, DIC images were collected every 10 s at 23°C. Z-stacks were captured soon after time-lapse imaging.

## Imaging of fixed oocytes

Fluorescence imaging was performed essentially as described in previous studies (Strickland et al., 2004; Burdyniuk et al., 2018), and the details are described below. To fix oocytes, 1-MA–stimulated oocytes in ASW or modified ASW were collected, placed in fixative solution (100 mM Hepes, pH 7.0, 50 mM EGTA, 10 mM MgSO$_4$, 0.5% Triton X-100, 1% formaldehyde, and 0.1% glutaraldehyde) in microtubes, and incubated for 1 h at RT with gentle rotation. Next, the fixative solution was replaced with PBS containing 0.2% Triton X-100, followed by an additional overnight incubation at 4°C with gentle rotation. The oocytes were washed three times with detergent-free PBS, and the PBS was then replaced with PBS containing 0.1%

borohydride. The microtubes containing oocytes were uncapped and allowed to stand at RT for 1.5–3 h. Thereafter, the oocytes were washed once with PBS and once with PBS containing 0.1% Triton X-100 (PBT); incubated in blocking solution (PBT containing 0.1% BSA and 5% goat serum) at RT for 3 h with gentle rotation; washed three times with PBT; absorbed onto poly-L-lysine (mol wt: 70,000–150,000 [Sigma-Aldrich])–coated coverslips; treated with Image-iT FX signal enhancer (Thermo Fisher Scientific) for 30 min at RT to block nonspecific binding of the fluorescent probes; washed three times with PBT; treated with an anti-α-tubulin monoclonal antibody (1:1400 YL1/2; AbD Serotec) diluted in PBS containing 0.5% Triton X-100 and 1% BSA (antibody diluent) at RT for 1.5 h; washed three times with PBT; treated with Alexa Fluor 488–conjugated secondary antibody diluted in antibody diluent (1:600 Alexa Fluor 488–conjugated anti-rat IgG and IgM antibody [Thermo Fisher Scientific] at RT for 1.5 h; washed three times with PBT; treated with phalloidin staining solution (PBT containing 100 nM Acti-stain 555 Fluorescent Phalloidin [Cytoskeleton]) at RT for 1 h; rinsed once with PBT; washed three times with PBS; and mounted on glass slides with ProLong Gold antifade reagent containing DAPI (Thermo Fisher Scientific). For mounting, the coverslips carrying the fixed oocytes were affixed to glass slides with double layers of double-sided adhesive tape (Scotch; 3M Company). Single slice images or Z-stacks (Z step of 1.0 µm) of the fixed samples were acquired on Carl Zeiss LMS 700 and 780 confocal microscopes equipped with 63× Plan-Apochromat 1.4 NA oil-immersion objective lenses. The images were adjusted for brightness and contrast and projected (maximum-intensity projection) using Zen2012 SP1 (black edition) for the LSM 700 and Zen2012 SP5 (black edition) for the LSM 780.

## Histone H1 kinase assay

To lyse oocytes, 6 µl of lysis buffer (80 mM β-glycerophosphate, 15 mM MgCl$_2$, 20 mM EGTA, pH 7.5, 100 mM sucrose, 100 mM KCl, 1 mM DTT, cOmplete EDTA free, 0.5% NP-40, and 1% phosphatase inhibitor cocktails 1 and 2) were added to 20 oocytes in 4 µl of ASW or modified ASW. The oocytes were incubated for 30 min on ice, followed by flash gentle vortexing and centrifugation (15 krpm, 15 min). The supernatant was analyzed via immunoblotting and used in the H1 kinase assays. For the immunoblotting, 2.5 µl (representing five oocytes) of supernatant was mixed with 2.5 µl of 2× LSB, followed by heating at 95°C for 5 min. For the H1 kinase assay, 0.5 µl of supernatant (representing an oocytes) was mixed on ice with 9.5 µl of reaction buffer (80 mM β-glycerophosphate, 15 mM MgCl$_2$, 20 mM EGTA, pH 7.5, 1 mM DTT, and cOmplete EDTA free) containing 3 µg of histone H1 (Roche) and 74 kBq of [γ-$^{32}$P] ATP (PerkinElmer). For the phosphorylation reaction, the mixture was incubated at 25°C for 15 min. The reaction was stopped by adding 10 µl of 2× LSB, followed by heating at 95°C for 5 min. 10 µl of each sample was loaded on a 12.5% SDS-PAGE gel. After SDS-PAGE, histone H1 phosphorylation was detected via autoradiography and quantified via liquid scintillation counting. For the autoradiography, the gels were exposed to imaging plates, and images were obtained with a Typhoon FLA 7000 (GE Healthcare). For the liquid scintillation counting, the gels were

subjected to Coommassie brilliant blue staining. The histone H1 bands were cut out from the gel and placed into separate microtubes. The microtubes were inserted into glass vials, and the radioactivity was quantified using a Tri Carb 3110 TR liquid scintillation counter (PerkinElmer).

### Other software
Adobe Photoshop was used to adjust brightness and contrast of all images. Adobe Illustrator was used to generate all figures. Adobe Premiere Pro was used to edit and compress all videos. Microsoft Excel was used to generate all graphs.

### Online supplemental material
Fig. S1 A shows $pH_i$ measurements in oocytes after 1-MA stimulation in the presence or absence of roscovitine. Fig. S1 B shows $pH_i$ measurements in oocytes after 1-MA stimulation in modified ASW for $pH_i$ clamping. Fig. S2 shows H1 kinase assays in oocytes after 1-MA stimulation at various clamped $pH_i$ values. Fig. S3 shows the GVBD morphologies at various clamped $pH_i$ values. Fig. S4 shows Z-stacks of oocytes to ascertain whether cytoplasmic granule invasion was completed at various clamped $pH_i$ values. Fig. S5 shows example data to explain the method used for $pH_i$ measurement. Videos 1, 2, 3, and 4 are time-lapse videos of GVBD in oocytes in ASW (Video 1) and in modified ASW for $pH_i$ clamping at ∼6.7 (Video 2), 7.0 (Video 3), and 7.2 (Video 4).

## Acknowledgments
We thank M. Matsushita (Ochanomizu University, Tokyo, Japan) for raising the anti-sfSGK-ΔN50 antibody, M. Terasaki (UConn Health, Farmington, CT) for providing vector pSP64-S, and E. Okumura (Tokyo Institute of Technology, Yokohama, Japan) for providing anti-sfCdc25, anti-sfAkt C-terminal fragment, and anti-starfish cyclin B antibodies. We also thank K. Yura (Ochanomizu University, Tokyo, Japan) for his contribution to the SGK isoform alignment.

This work was supported by Japan Society for the Promotion of Science KAKENHI grant no. 17K07405, the Takeda Science Foundation, and the Cooperative Program provided by the Atmosphere and Ocean Research Institute, University of Tokyo.

The authors declare no competing financial interests.

Author contributions: conceptualization, E. Hosoda, D. Hiraoka, and K. Chiba; data curation, E. Hosoda; formal analysis, E. Hosoda; funding acquisition, K. Chiba; investigation, E. Hosoda and D. Hiraoka; investigation (cloning of cDNA encoding sfSGK), S. Omi and N. Hirohashi; methodology, E. Hosoda, D. Hiraoka, and K. Chiba; project administration, E. Hosoda, D. Hiraoka, and K. Chiba; resources, E. Hosoda, D. Hiraoka, T. Kishimoto, and K. Chiba; supervision, D. Hiraoka and K. Chiba; validation, E. Hosoda and D. Hiraoka; visualization, E. Hosoda, D. Hiraoka, and K. Chiba; writing (original draft), E. Hosoda and D. Hiraoka; and writing (review and editing), all authors.

Submitted: 21 December 2018

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
