## [Reviewer comments · The Journal of Cell Biology]

SGK regulates pH increase and cyclin B-Cdk1 activation to resume meiosis in starfish ovarian oocytes

Enako Hosoda, Daisaku Hiraoka, Noritaka Hirohashi, Saki Omi, Takeo Kishimoto, and Kazuyoshi Chiba

Corresponding Author(s): Kazuyoshi Chiba, Ochanomizu University and Enako Hosoda, Ochanomizu University

Review Timeline:	Submission Date:	2018-12-21
	Editorial Decision:	2019-02-04
	Revision Received:	2019-07-19
	Editorial Decision:	2019-08-07
	Revision Received:	2019-08-11

Monitoring Editor: Jan Ellenberg

Scientific Editor: Tim Spencer

Transaction Report:

DOI: <https://doi.org/N/A>

February 4, 2019

Re: JCB manuscript #201812133

Prof. Kazuyoshi Chiba
Ochanomizu University
Department of Biological Sciences
2-1-1 Ohtsuka,
Bunkyo-ku,, Tokyo 112-8610
Japan

Dear Prof. Chiba,

Thank you for submitting your manuscript entitled "SGK regulates pH increase and cyclin B-Cdk1 activation to resume meiosis in starfish ovarian oocytes". We apologize for the delay in providing you with a decision on your paper. In any case, the manuscript has now been assessed by expert reviewers, whose comments are appended to this letter. We invite you to submit a revision if you can address the reviewers' key concerns, as outlined here.

You will see that all three reviewers find the study to be convincing and interesting for the JCB audience. However, reviewers #1 and 2 have raised a few concerns that will need to be addressed before the paper would be deemed suitable for publication, including the need for supplemental analyses to confirm your pH measurements (reviewer #1), better characterization of the stage where GVBD arrests (rev#2), and further corroboration for the claim that blocking pH changes interferes with nuclear envelope/germinal vesicle breakdown (rev#1). We hope that you will be able to address these, and each of the other reviewer comments, in full in a revised manuscript.

GENERAL GUIDELINES:

Text limits: Character count for an Article is < 40,000, not including spaces. Count includes title page, abstract, introduction, results, discussion, acknowledgments, and figure legends. Count does not include materials and methods, references, tables, or supplemental legends.

Figures: Articles may have up to 10 main text figures. Figures must be prepared according to the policies outlined in our Instructions to Authors, under Data Presentation, <http://jcb.rupress.org/site/misc/ifora.xhtml>. All figures in accepted manuscripts will be screened prior to publication.

IMPORTANT: It is JCB policy that if requested, original data images must be made available. Failure to provide original images upon request will result in unavoidable delays in publication. Please ensure that you have access to all original microscopy and blot data images before submitting your revision.

Supplemental information: There are strict limits on the allowable amount of supplemental data. Articles may have up to 5 supplemental figures. Up to 10 supplemental videos or flash animations are allowed. A summary of all supplemental material should appear at the end of the Materials and methods section.

The typical timeframe for revisions is three months; if submitted within this timeframe, novelty will not be reassessed at the final decision. Please note that papers are generally considered through only one revision cycle, so any revised manuscript will likely be either accepted or rejected.

Thank you for this interesting contribution to Journal of Cell Biology. You can contact us at the journal office with any questions, cellbio@rockefeller.edu or call (212) 327-8588.

Sincerely,

Jan Ellenberg, PhD
Monitoring Editor
JCB

Tim Spencer, PhD
Deputy Editor
Journal of Cell Biology
ORCID: 0000-0003-0716-9936

Reviewer #1 (Comments to the Authors (Required)):

The manuscript by Hosoda and colleagues identifies SGK (serum- and glucocorticoid-regulated kinase) as a key component in the signaling leading to meiotic maturation in starfish oocytes. They show that on the one hand, SGK is required for triggering the cell cycle by activation of cdk1-cyclin B. On the other hand, SGK controls the change in intracellular pH. In previous work, the Chiba lab showed that intracellular pH changes have important functions in controlling the MI arrest required to coordinate spawning and fertilization with meiotic progression in starfish oocytes. Now they show that there is a pH change coinciding with early steps of meiotic maturation, and that this pH change is required for completion of NEBD.

The main novelties are: (i) the identification of SGK as a new regulator of meiotic maturation in starfish, and (ii) showing that intracellular pH is tightly controlled during maturation, and that these changes in pH have important regulatory functions. In my opinion, this latter point renders the manuscript particularly interesting for the broader readership of the Journal of Cell Biology, because it is not commonly considered that intracellular pH is undergoing rapid and controlled changes, and that it would have major roles in controlling processes such as nuclear envelope breakdown or

cytoskeletal dynamics. This may be relevant in oocytes in other species, and possibly in other cell types as well.

For these reasons, I find the manuscript in principle suitable for publication in the Journal of Cell Biology. The manuscript is also well written, the figures are very clear, and the data shown supports the conclusions drawn. However, I would have two major requests for additional data, which in my opinion is necessary to make the manuscript publishable:

1. As detailed above, I find changes in intracellular pH one of the key conclusions. Therefore, it would be critical to better document and extend these measurements as follows: (i) there are many references made to previous publications, some of which are in not easily accessible journals. The authors have to make sure that all necessary details are contained in the present manuscript (in the methods section) regarding how pH is measured and manipulated. (ii) Specifically, how are BCECF-dextran measurements calibrated? How reliable are they? It would also be helpful to show the fluorescent images, and provide a description how they were quantified. (iii) The authors also use buffered artificial sea water to manipulate intracellular pH and claim that in these oocytes pH is constant. It would be critical to confirm these by imaging BCECF-dextran, which would also report on the actual value on the intracellular pH, and further confirm the reliability of BCECF-dextran measurements.

2. Another key finding is that blocking pH change prevents complete nuclear envelope breakdown. This needs to be better explored than showing transmitted light images. Staining of DNA in fixed samples would already provide critical details whether chromosomes traveled to the animal pole, whether there is aneuploidy resulting from these defects, and by adding sperm at different times it would be interesting to see when block-of-polyspermy is activated. Co-staining with phalloidin would provide additional details with regard to the various actin structures that are required for nuclear envelope breakdown and chromosome transport. Furthermore, it would be interesting to see whether the spindle forms normally by an anti-tubulin antibody staining.

Reviewer #2 (Comments to the Authors (Required)):

With this manuscript, Dr. Hosoda and colleagues have investigated the signaling pathway involved in G2/M transition and MI progression in starfish oocytes after methyladenine stimulation. The authors have cloned the star fish SGK kinase (sfSGK) and generated antibodies against a protein devoid of the amino terminus of the (anti-sfSGK- Δ N50) and one against the TM domain which is target for Torc2 phosphorylation. In addition, they use a commercially available antibody anti human antibody that recognizes the A loop phosphorylation (that they term sfSGK-pT312 (A-loop)). Using these antibodies, they document a shift in the sfSGK that coincides with the phosphorylation of the two sites. They also show phosphorylation of the sfSGK prior to Cdk1 and activation. Using different inhibitors, they conclude that Torc2 and PDK1 are responsible for the phosphorylation. Using a more physiological experimental setting, they document that SGK activation occurs also in oocytes exposed to MeA prior to spawning. They provide evidence that SGK activation is necessary for the shift in pHi of the oocytes and Cdk1 activation. The shift in pHi is necessary for completion of GVBD.

The study is thorough and convincingly links SGK activation to changes in pHi in the starfish oocytes. Perhaps, a better morphological characterization of the stage of GVBD in which the oocytes are arrested at low pHi would have strengthen the study.

1. Since this and the accompanying paper will be likely read at the same time, it would be helpful to the reader if the use the same nomenclature for the antibodies used in the two studies.
2. Fig 2C Why so much variation in the loading among different lanes? The authors could include a WB with more even loading at the times used. In the discussion of these data in the results (line 154-157) the authors note a shifted band (panel 2C) that is not recognized by the pA loop antibody and they propose that the shift is due to phosphorylation at the HM site. To confirm that the shifted band not recognized by the A loop antibody is indeed the product of Torch2-dependent phosphorylation of the HM, the authors could have used the antibody developed by their colleagues and used in the accompanying manuscript.
3. Fig.5 The authors report that clamping pHi at pH 6.7 causes a delay in oocyte GVBD (Fig.5B). However, in Fig. 5A, they show that Cdk1-pY15 phosphorylation disappears at 4 min at pHi of 6.7 while this occurs later at pHi 7.0. Similarly, the Cdc25 shift is delayed at pHi of 7.0. How do they reconcile this discrepancy between the phosphorylation pattern and GVBD?
4. Effect of pHi in completion of GVBD. The authors claim that completion of GVBD is impaired if oocyte pHi remains at 6.7. Have the authors considered the possibility that Cdk1 activation at pH6.7 is not sustained or not at levels sufficient to complete NEBD? Direct measurements of CDK activity could help addressing this issue. Also the movies included show a defect in cytoplasm invasion of the GV at pH 6.7. Are there other morphological correlates (e.g. IFF markers) that can be used to define the stage in which the oocytes are blocked? Would DNA staining show differences in condensation of the chromosomes?
5. Throughout the manuscript the format of citations should be revised.
6. Line 23. fertilizability?

Reviewer #3 (Comments to the Authors (Required)):

Overall appreciation

The article by Hosada et al explores the role played by SGK (serum and glucocorticoid-regulated kinase) during oocyte maturation in starfish, rather than Akt/PKB pathway as previously reported. The authors convincingly demonstrate that following 1-methyl adenine induced maturation, that SGK is activated via a two-step process: first the C-terminal HM motif is phosphorylated by TORC2A-loop allowing interaction between SGK and PDK1 which phosphorylates the A-loop causing SGK activation. Both PDK1 and TORC2 activation require PI3 kinase upstream activity. This causes germinal vesicle breakdown and is associated with two separate events: a pH increase from pH6.7 (the pH inside starfish ovary) to pH7.2 (in sea water) or pH7.0 (if the oocyte remains in the ovary) and activation of Cdk1 via SGK-induced phosphorylation of Cdc25 (activating phosphorylation) and Myt1 (inhibitory phosphorylation). Interestingly, preventing the pH increase delays GV breakdown but not the activation of Cdk1. Furthermore, the pH increase is required for the migration of granules into the GV space following breakdown. Finally, since SGK activation and phosphorylation on the A loop peaks then reduces following GV breakdown, an unidentified SGK A loop phosphatase is proposed to play a crucial role in limiting the pH rise in oocytes within the ovary to pH7.0: this may prolong meiosis I thereby ensuring monospermy. This is a very interesting, thorough and convincing demonstration of the molecular events leading to oocyte maturation by a team that has led the way in cell cycle regulation of oocyte maturation. I am very positive about this

manuscript.

Detailed summary of results and comments

Reagents

Antibodies generated to starfish SGK3:

□N50 Ab to SGK lacking N-ter 50aa (retains Thr312 PDK1 phosph motif in A loop)

HM Ab to SGK 17aa C-ter peptide containing hydrophobic motif (site of Ser479 mTORC2 phosphory)

Both antibodies detected 56kDa protein, and detected mobility shift after 1MA

sfSGK-p312 A-loop Ab: Anti Phospho Human antibody to A loop phospho SGK: band detected in 1MA oocytes but not before 1MA. Band also 56kDa

Confirmed cross reactivity of HsAb by using purified starfish SGK and probing with HsAb.

All following figures strongly support the scientific claims.

Fig. 1. Sequence alignment and antibody cross-reactivity.

Fig. 2. Kinetics: After 1MA sfSGK is activated 1 min., Cdc25 7 min., Cdk1 at 10 min. and GVBD 17 min. in isolated oocytes

Mobility shift and A loop phospho. detected 1 min. after 1MA, peaked at 3 min.

Cdc25 hyper phosph. at 7 min. (Fig. 2a)

Cdk1 Tyr15 dephosph. at 10 min. (Fig. 2a)

- A loop Ab used for Thr312 of SGK

- Checked Ser477, of Akt by TORC2 phospho. Similar to SGK

- Checked ovary epithelium, no antibody signal detected (Fig. 2b)

In situ kinetics of ovary intact oocytes by injecting 1MA into body cavity (similar to isolated oocytes)

GVBD in 25 min. after 1MA (spawning at 30 min.) (Fig. 2c)

SGK and Cdc25 activated within 5 min, Cdk1 within 25 min.

Involvement of PDK1 and TORC2

A loop phosph Thr312: inhibited by BX795 (PDK1 inh). (Fig. 2d)

HM phosph Ser479: inhibited by pp242 (a TOR catalytic subunit inh). pp242 also inh. A loop phosph (cryptic site) (Fig. 2d)

Mobility shift: inhibited by pp242 but not BX795. (Fig. 2e)

Thus, shift caused by HM phosph. by TORC2.

Rapamycin had no effect on phospho, thus ruling out TORC1 (which also contains TOR cat. subunit) (Fig. 2e)

Mobility shift indicated that upper migrating band was the active dual phosphorylated form.

Lower band = non phospho

Middle band = Ser479, TORC2 phospho band (pp242 inh)

Upper band = Thr312, PDK1 and Ser479, TORC2 phospho band: Fig. 2e

First, only upper band was absent following BX which targets Thr312. Fig. 2d, lane 10. Fig. 2f, lane 5.

Second, upper and middle bands absent following pp242 which targets Thr479, Fig. 2d, lane 6. Fig. 2f, lane 6.

Third, upper and middle band were absent following wortmanin (PI3K inh). Fig. 2d, lane 4. Fig. 2f, lane 4. Some residual Phosph. Of middle band.

Mobility shift not inhibited by BX, Fig.2d, lane 10.

Thus, TORC2 Thr479 phospho. necessary for PDK1 Ser312 phospho. Both require PI3K

Fig. 3. SGK activation necessary for pH increase

Anti-SGK HM blocked mobility shift of SGK following microinjection and 1MA - Fig. 3a
Anti-SGK HM also blocked pH increase

Fig. 4. What causes Cdk1 activation - pH increase or SGK activation?

Previously GVBD occurred even when pH increase was blocked.

However, Anti-SGK HM blocked GVBD

Anti-SGK HM blocked Cdc25 activation and Cdk1 Tyr dephosphorylation/activation. Fig. 4c.

Rescue of Anti-SGK HM with mRNA encoding SGK-T479E, 1MA now caused Cdk1 activation and GVBD - Fig. 4

Also, the pH increase and Cdk1 activation are independent events.

Fig. 5. Reduced pH increase delays GVBD and granule invasion of GV but does not affect time of Cdk1 activation.

Cdk1 activation kinetics independent of pH increase.

Supp. Fig. 1. pH increase following 1 MeAd is not blocked by Cdk1 kinase inhibitor roscovitine

Supp. Fig. 2. GV envelope morphology at different pHs.

Supp. Fig. 3. Low pH delays complete invasion of granules into GV space following breakdown.

Movie 1. Beautiful GVBD in ASW. Granule invasion complete at 30min, but scale bar is missing.

Movie 2. Beautiful GVBD with reduced granule invasion (at 1hr20min) at pH6.7, but scale bar is missing.

Movie 3. Beautiful GVBD with reduced granule invasion (at 1hr10min) at pH7.0, but scale bar is missing.

Movie 4. Beautiful GVBD with granule invasion at pH7.2 (at 40-50min), but scale bar is missing.

Additional comments

Interesting to note that the pH increase induced by SGK is damped and remains at pH7.0 so that oocytes enter MI arrest induced by pH7.0, thus leading to monospermy. This is associated with dephosphorylation of the A loop by an as yet unknown phosphatase following Cdk1 activation (Fig. 5a)

It is also interesting to note that SGK phosphorylates Cdc25 (activating) and Myt1 (inhibiting) thus promoting Cdk1 Tyr 15 dephosphorylation (activation).

JCB manuscript #201812133

(Hosoda et al., SGK regulates pH increase and cyclin B-Cdk1 activation to resume meiosis in starfish ovarian oocytes)

Response to reviewers' comments and suggestions

First of all, we would like to thank all of three reviewers for their constructive comments and suggestions. Below, the review comments are pasted with blue, and our responses are described with black. We hope that these responses will meet with approval of all the reviewers.

Reviewer #1

(Comments to the Authors (Required)):

The manuscript by Hosoda and colleagues identifies SGK (serum- and glucocorticoid-regulated kinase) as a key component in the signaling leading to meiotic maturation in starfish oocytes. They show that on the one hand, SGK is required for triggering the cell cycle by activation of cdk1-cyclin B. On the other hand, SGK controls the change in intracellular pH. In previous work, the Chiba lab showed that intracellular pH changes have important functions in controlling the MI arrest required to coordinate spawning and fertilization with meiotic progression in starfish oocytes. Now they show that there is a pH change coinciding with early steps of meiotic maturation, and that this pH change is required for completion of NEBD.

The main novelties are: (i) the identification of SGK as a new regulator of meiotic maturation in starfish, and (ii) showing that intracellular pH is tightly controlled during maturation, and that these changes in pH have important regulatory functions. In my opinion, this latter point renders the manuscript particularly interesting for the broader readership of the Journal of Cell Biology, because it is not commonly considered that intracellular pH is undergoing rapid and controlled changes, and that it would have major roles in controlling processes such as nuclear envelope breakdown or cytoskeletal dynamics. This may be relevant in oocytes in other species, and possibly in other cell types as well.

For these reasons, I find the manuscript in principle suitable for publication in the Journal of Cell Biology. The manuscript is also well written, the figures are very clear, and the data shown supports the conclusions drawn. However, I would have two major requests for additional data,

which in my opinion is necessary to make the manuscript publishable:

1. As detailed above, I find changes in intracellular pH one of the key conclusions. Therefore, it would be critical to better document and extend these measurements as follows: (i) there are many references made to previous publications, some of which are in nor easily accessible journals. The authors have to make sure that all necessary details are contained in the present manuscript (in the methods section) regarding how pH is measured and manipulated. (ii) Specifically, how are BCECF-dextran measurements calibrated? How reliable are they? It would also be helpful to show the fluorescent images, and provide a description how they were quantified.

As the response to issues "i" and "ii", we have added a detailed description about method for intracellular pH (pH_i) measurement in the Method section (pages 25-27, lanes 590-640). We have also added Fig. S5 showing examples of the measurement to help readers understand the method.

In the measurement, we used pH-dependent fluorescent ratio of injected BCECF-dextran between fluorescent intensities obtained by 436 nm excitation and 495 nm excitation. A higher pH gives a higher ratio. To measure fluorescence intensity and ratio, we defined region of interest (ROI) on the fluorescence image of each oocyte using HCImage software. The software automatically calculated an averaged intensity and the ratio in each ROI.

We can determine steady state pH values in unstimulated oocytes by injecting standard solutions, which have various known pH values, into BCECF-injected oocytes. The fluorescent ratio changed when the pH_i was different from that of injected standard solution. When the pH_i was almost equal to that of injected standard solution, the ratio did not change. In this method, pH_i differences as little as 0.05 are detectable.

This method is not suitable for time course measurement because several times of injections are required to determine pH_i values. Such multiple injections at each time point are unrealistic. So, we took advantage of two principles. The one is based on the steady state calibration: when oocytes are incubated in sodium-free ASW containing $\text{CH}_3\text{COONH}_4$ (modified ASW), the steady state pH_i value of the oocytes becomes approximately 0.2 unit higher on average than pH of the modified ASW (Moriwaki et al., 2013). The other is that there is a linear correlation between the intensity ratio and pH_i values at least in a pH_i range of 6.6 to 7.5 (the data showing this linear correlation have not been published anywhere, and therefore we have added it in Fig. S5). For measurement of time course change in pH_i values, we performed a standard measurement in each experiment. In the standard, we adjusted pH_i values of oocytes at ~ 6.6 and ~ 7.5 using the modified ASW, measured the fluorescent ratio, and plotted the each ratio against each pH_i values (6.6 and 7.5). This plot gave a standard linear function

corresponding to a line connecting the two points on the plot. We then performed the time-lapse recordings of the fluorescence ratio under the experimental conditions (e.g. 1-MA treatment). We finally apply the ratio at each time point to the standard linear function, thereby converted the ratios to the pH_i values. We have added Fig. S5 showing example data to explain the method for pH_i measurement: Fig. S5A shows example fluorescence images of the oocytes with ROI; Fig. S5B shows an example measurement of a steady state pH_i by injecting pH-standard solutions; Fig S5C is a plot showing a linear correlation between pH_i and the fluorescence ratio; Fig S5D shows example ratio measurements in the oocytes at clamped pH_i values of ~ 6.6 and ~ 7.5 ; Fig S5E shows a plot to calculate the standard linear function; Fig. S5F shows example graphs of a time-lapse recording of the fluorescent ratio in 1-MA-stimulated oocytes, and pH_i values converted from the ratio. We used a pH meter which has a resolution of 0.001 pH unit to adjust pH of ASW, the modified ASW, and the pH-standard solutions.

Moriwaki, K., et al. 2013. Arrest at metaphase of meiosis-I in starfish oocytes in the ovary is maintained by high CO_2 and low O_2 concentrations in extracellular fluid. *Zoolog. Sci.* 31:975–984.

(iii) The authors also use buffered artificial sea water to manipulate intracellular pH and claim that in these oocytes pH is constant. It would be critical to confirm these by imaging BCECF-dextran, which would also report on the actual value on the intracellular pH, and further confirm the reliability of BCECF-dextran measurements.

We have added Fig. S1B showing a measurement of pH_i in the oocytes treated with 1-MA in modified ASW for clamping pH_i . The measurement confirmed that pH_i values in unstimulated oocytes were changed depending on pH of the modified ASW, and did not increase after 1-MA stimulation. We have added description about this result in Results (pages 11, lines 259-262).

2. Another key finding is that blocking pH change prevents complete nuclear envelope breakdown. This needs to be better explored than showing transmitted light images. Staining of DNA in fixed samples would already provide critical details whether chromosomes traveled to the animal pole, whether there is aneuploidy resulting from these defects, and by adding sperm at different times it would be interesting to see when block-of-polyspermy is activated. Co-staining with phalloidin would provide additional details with regard to the various actin structures that

are required for nuclear envelope breakdown and chromosome transport. Furthermore, it would be interesting to see whether the spindle forms normally by an anti-tubulin antibody staining.

We have added Fig. 6 and 7 showing fluorescence imaging of fixed oocytes.

We performed co-staining of F-actin, microtubules and chromosomes in pH_i -clamped fixed oocytes. Lenart Lab previously found dynamic changes in F-actin and microtubule architectures during a period from GVBD to spindle formation as summarized below.

Just before GVBD, an F-actin shell essential for nuclear envelope (NE) fragmentation forms on the inner surface of the GV (Lénárt et al., 2005; Mori et al., 2014). The shell disappears within 1 min after GVBD simultaneously with formation of F-actin meshwork in entire inner GV area and of F-actin patches surrounding chromosomes (Lénárt et al., 2005). At this time, chromosomes are randomly scattered in inner GV area, but thereafter are transported by contractile flow of the actin meshwork toward the animal pole where two centrosomes exist near plasma membrane. (Lénárt et al., 2005; Mori et al., 2011; Bun et al., 2018). Subsequently, the actin patches are disappeared, and the transported chromosomes are captured by astral microtubules from centrosomes, followed by formation of a spindle with aligned chromosomes at the animal pole (Lénárt et al., 2005; Burdyniuk et al., 2018).

In our observations, transient F-actin shell, subsequent F-actin meshwork and patches were formed at all of the clamped pH_i values. Spindles formed in the oocytes when pH_i was clamped at 7.2 and 7.0. However, in oocytes at a clamped pH_i of 6.7, two asters were observed, but they did not form a spindle even after prolonged incubation. These oocytes still had F-actin meshwork in inner GV area even after prolonged incubation, and some chromosomes were not gathered to the animal pole. Thus, we concluded that formation of F-actin meshwork is pH_i -independent, and that chromosome transport driven by the meshwork and microtubule organization for spindle formation are perturbed at a low pH_i . We have added description about these defects in Results (pages 13-14, lines 291-335) and Discussion (pages 16-17, lines, 377-392).

We discussed in the manuscript of initial submission that a cause of defect in invasion of cytoplasmic granules into GV region at pH_i 6.7 may be less efficient NE fragmentation, and that formation of F-actin shell, which is required for efficient NE fragmentation, may be disrupted at pH_i 6.7. In the experiments for revision, our staining showed that F-actin shell was not disturbed even when pH_i was clamped at 6.7. Lenart Lab proposed that small F-actin protrusions formed on the shell is essential for efficient NE fragmentation (Mori et al., 2014; Wesolowska et al., 2018 *preprint*). Thus, protrusions on the shell may be disrupted at pH_i of 6.7. Unfortunately, the protrusions were not visible on our imaging probably due to insufficient resolution. Thus we

have added description about protrusions in Discussion as an issue to be investigated in future study (pages 16, lines 369-376).

Lenart Lab described actin meshwork formation and subsequent chromosome transport as events that occur 'after GVBD'. We follow their papers and describe these events as those 'after GVBD' in our revised manuscript. However, we had used a term of "completion of GVBD" to refer to completion of cytoplasmic granule invasion. This is definitely confusing for readers because the granule invasion proceeds during a period of actin meshwork-dependent chromosome transport (chromosome transport 'after GVBD' occurs during a period in which granule invasion proceeds toward 'completion of GVBD'). Thus we decided not to use 'completion of GVBD'. Instead, we described GVBD as a moment at which the granule invasion starts, and progression of the granule invasion as an event after GVBD (chromosome transport and granule invasion proceed after GVBD) in the revised manuscript.

On the basis of these observations, we have revised a model of SGK-dependent meiotic resumption in ovarian oocytes. So far it is still unclear whether incompleting granule invasion causes serious consequences in oocyte maturation and/or following development, our finding of defects in chromosome gathering and spindle formation at a low pH_i further emphasizes importance of pH_i increase in ovarian oocytes. So, in the revised model, we describe that sfSGK-dependent pH_i increase is a prerequisite for actin-driven chromosome transport and microtubule organization for spindle formation in ovarian oocytes. We have added Fig 8 and description about this model (pages 15, lines 349-356).

In addition, we also found that unaligned chromosomes in most of the spindles formed at a clamped pH_i of 7.0. This pH_i value is similar to the estimated pH_i of 1-MA-stimulated ovarian oocytes (Moriwaki et al., 2013). We previously claimed that ovarian oocytes undergo a secondary arrest mainly at metaphase-I because majority of oocytes (~70%) soon after spawning were at metaphase-I (Harada et al., 2003). In this observation by Harada et al., remaining ~30% of the oocytes soon after spawning were still at prometaphase. Considering the presence of unaligned chromosomes at a clamped pH_i of ~7.0, more ovarian oocytes may stay at late prometaphase than at metaphase; 'MI-arrested starfish oocytes' seem to include oocytes at late prometaphase and metaphase. We also speculate that pH_i ~7.0 might be somewhat inadequate condition for the establishment and/or maintenance of bi-oriented attachment. If this is the case, anaphase onset before spawning might increase probability of aneuploidy, and therefore the arrest at stages before anaphase onset would protect oocytes from this potential risk of aneuploidy. Although it remains to be determined which is the actual main stage of the MI arrest, we think that an important feature could be that oocytes wait for spawning at stages

before onset of anaphase-I. We have added description about these issues in Results (pages 13-14, lines 312-315) and Discussion (pages 17, lines 393-402; pages 18, lines 417-422).

A suggested experiment of adding sperms to pH_i clamped oocytes is very interesting, but unfortunately it was unsuccessful, because sperms did not swim in the modified ASW. So, we can hardly distinguished effects of pH_i alteration on fertilization from those by interrupted sperm mobility. So far, we do not have method to stably control pH_i values without the modified ASW, we would like to investigate this issue in a future study.

Bun et al. 2018. A disassembly-driven mechanism explains F-actin-mediated chromosome transport in starfish oocytes. *eLife*. 7:e31469. doi:10.7554/eLife.31469.

Burdyniuk et al. 2018. F-Actin nucleated on chromosomes coordinates their capture by microtubules in oocyte meiosis. *J. Cell Biol.* 2018 217:2661-2674.

Harada, K., E. Oita, and K. Chiba. 2003. Metaphase I arrest of starfish oocytes induced via the MAP kinase pathway is released by an increase of intracellular pH. *Development*. 130:4581–4586.

Lénárt et al. 2005. A contractile nuclear actin network drives chromosome congression in oocytes. *Nature*. 436:812-818.

Mori et al. 2011. Intracellular transport by an anchored homogeneously contracting F-actin meshwork. *Curr. Biol.* 21:606-611.

Mori et al. 2014. An Arp2/3 nucleated F-actin shell fragments nuclear membranes at nuclear envelope breakdown in starfish oocytes. *Curr. Bio.* 24:1421–1428.

Wesolowska et al. 2018. An F-actin shell ruptures the nuclear envelope by sorting pore-dense and pore-free membranes in meiosis of starfish oocytes. *bioRxiv*. doi.org/10.1101/480434 (Preprint posted November 28, 2018).

Reviewer #2

(Comments to the Authors (Required)):

With this manuscript, Dr. Hosoda and colleagues have investigated the signaling pathway involved in G2/M transition and MI progression in starfish oocytes after methyladenine stimulation. The authors have cloned the star fish SGK kinase (sfSGK) and generated antibodies against a protein devoid of the amino terminus of the (anti-sfSGK- Δ N50) and one against the TM domain which is target for Torch2 phosphorylation. In addition, they use a commercially available antibody anti human antibody that recognizes the A loop phosphorylation (that they term sfSGK-pT312 (A-loop). Using these antibodies, they document a shift in the sfSGK that coincides with the phosphorylation of the two sites. They also show phosphorylation of the sfSGK prior to Cdk1 and activation. Using different inhibitors, they conclude that Torc2 and PDK1 are responsible for the phosphorylation. Using a more physiological experimental setting, they document that SGK activation occurs also in oocytes exposed to MeA prior to spawning. They provide evidence that SGK activation is necessary for the shift in pHi of the oocytes and Cdk1 activation. The shift in pHi is necessary for completion of GVBD.

The study is thorough and convincingly links SGK activation to changes in pHi in the starfish oocytes. Perhaps, a better morphological characterization of the stage of GVBD in which the oocytes are arrested at low pHi would have strengthen the study.

1. Since this and the accompanying paper will be likely read at the same time, it would be helpful to the reader if the use the same nomenclature for the antibodies used in the two studies.

According to the suggestion, we have revised antibody nomenclatures as follows.

In the manuscript by Hosoda et al.

anti-sfAkt C-terminus -> anti-sfAkt C-terminal fragment

anti-starfish phospho-Akt (Ser477) -> anti-sfAkt phospho-Ser477

anti-phospho-Tyr15 Cdk1 -> anti-Cdk1 phospho-Tyr15

In the manuscript by Hiraoka et al.

anti-sfSGK-C -> anti-sfSGK-HM

anti-phospho-sfAkt-Ser477 -> anti-sfAkt phospho-Ser477

anti-phospho-Cdk1-Tyr15 -> anti-Cdk1 phospho-Tyr15

2. Fig 2C Why so much variation in the loading among different lanes? The authors could include a WB with more even loading at the times used.

At the sampling of ovarian oocytes, we can not exactly count the number of oocytes in the collected ovary. Therefore, there is variation in amount of proteins among samples. Fortunately, we had frozen stocks of the residual samples after use to obtain the indicated data in Fig 2C. Using these samples, we checked the amounts of proteins in the samples, then properly diluted samples to load more even amounts. Then, we replaced the data with new one in which more even amounts of samples were loaded (Fig. 2C).

In the discussion of these data in the results (line 154-157) the authors note a shifted band (panel 2C) that is not recognized by the pA loop antibody and they propose that the shift is due to phosphorylation at the HM site. To confirm that the shifted band not recognized by the A loop antibody is indeed the product of Torch2-dependent phosphorylation of the HM, the authors could have used the antibody developed by their colleagues and used in the accompanying manuscript. So far, we have tried unsuccessfully to generate a phospho-specific antibody to detect HM phosphorylation of sfSGK. The antibody used in the accompanying paper is against phosphorylated HM of 'sfAkt', but not of 'sfSGK'. This phospho-sfAkt-HM antibody did not cross-react with sfSGK. Therefore we can not directly detect HM phosphorylation of sfSGK so far. However, using chemical inhibitors pp242 (TORC1/2 inhibitor) and rapamycin (TORC1 inhibitor), we showed that the shift is dependent on TORC2 activity (Fig. 2D). We believe that this observation supports our notion that the shift is due to TORC2-dependent HM phosphorylation.

3. Fig.5 The authors report that clamping pHi at pH 6.7 causes a delay in oocyte GVBD (Fig.5B). However, in Fig. 5A, they show that Cdk1-pY15 phosphorylation disappears at 4 min at pHi of 6.7 while this occurs later at pHi 7.0. Similarly, the Cdc25 shift is delayed at pHi of 7.0. How do they reconcile this discrepancy between the phosphorylation pattern and GVBD?

Our interpretation of these results is that signaling leading to cyclin B-Cdk1 activation prefers lower pHi whereas processes after cyclin B-Cdk1 toward GVBD prefers higher pHi. Cyclin B-Cdk1 is activated earlier at pHi 6.7 than at pHi 7.0, but processes from cyclin B-Cdk1 activation to GVBD takes a longer time at pHi 6.7 than that at pHi 7.0. When compared oocytes at pHi 6.7 with those at pHi 7.0, difference in time to cyclin B-Cdk1 activation is smaller than difference in

time from cyclin B-Cdk1 activation to GVBD; total time from 1-MA stimulation to GVBD is longer at pH_i 6.7 than that at pH_i 7.0. We had not mentioned about an acceleration in cyclin B-Cdk1 activation and Cdc25 hyperphosphorylation at pH_i 6.7 in the initial manuscript. So, we have added description about this issue in Results (pages 12, lanes 267-269) and Discussion (pages 15, lines 352-356) in the revised manuscript.

4. Effect of pH_i in completion of GVBD. The authors claim that completion of GVBD is impaired if oocyte pH_i remains at 6.7. Have the authors considered the possibility that Cdk1 activation at pH_i 6.7 is not sustained or not at levels sufficient to complete NEBD? Direct measurements of CDK activity could help addressing this issue.

According to the suggestion, we measured Cdk1 activity using histone H1 as a substrate. The activity in the oocytes at a clamped pH_i of 6.7 was maintained at a comparable level to those at clamped pH_i values of 7.0 and 7.2. Thus, levels of Cdk1 activity is not a cause of the defect observed at a clamped pH_i of 6.7. We have added Fig. S2 showing H1 kinase assay, and description about this issue (page 12, lines 271-272 and lines 279-284).

Also the movies included show a defect in cytoplasm invasion of the GV at pH_i 6.7. Are there other morphological correlates (e.g. IFF markers) that can be used to define the stage in which the oocytes are blocked? Would DNA staining show differences in condensation of the chromosomes?

We focused on F-actin, microtubules, and chromosomes to define the stage in which the oocytes are blocked. Previous studies by Lenart Lab. demonstrated that during a period of the cytoplasmic granule invasion, F-actin and microtubule architectures dynamically change to transport chromosomes and to form spindles as summarized below.

Just before GVBD, an F-actin shell essential for nuclear envelope (NE) fragmentation forms on the inner surface of the GV (Lénárt et al., 2005; Mori et al., 2014). The shell disappears within 1 min after GVBD simultaneously with formation of F-actin meshwork in entire inner GV area and of F-actin patches surrounding chromosomes (Lénárt et al., 2005). At this time, chromosomes are randomly scattered in inner GV area, but thereafter are transported by contractile flow of the actin meshwork toward the animal pole where two centrosomes exist near plasma membrane. (Lénárt et al., 2005; Mori et al., 2011; Bun et al., 2018). Subsequently, the actin patches are disappeared, and the transported chromosomes are captured by astral microtubules from centrosomes, followed by formation of a spindle with

aligned chromosomes at the animal pole (Lénárt et al., 2005; Burdyniuk et al., 2018).

We performed co-staining of F-actin, microtubules and chromosomes in pH_i -clamped fixed oocytes. In our observations, chromosomes were condensed as normal at all of the clamped pH_i values. Transient F-actin shell, subsequent F-actin meshwork and patches were formed at all clamped pH_i values. Spindles were formed in the oocytes when pH_i was clamped at 7.2 and 7.0. However, in oocytes at a clamped pH_i of 6.7, two asters were observed, but they did not form a spindle even after prolonged incubation. These oocytes still had F-actin meshwork in inner GV area even after prolonged incubation, and some chromosomes were not gathered to the animal pole. Thus, we concluded that at a clamped pH_i of 6.7, formation of F-actin meshwork occurs, but oocytes stalled at a subsequent stage of actin meshwork-driven chromosome transport. We have added Fig. 6 and 7 showing the fluorescence images, and description about these results in Results (page 13-14, lines 291-335) and Discussion (page 16-17, lines 377-402; page 18, lines 417-422).

We discussed in the manuscript of initial submission that a cause of defect in invasion of cytoplasmic granules into GV region at pH_i 6.7 may be an less efficient NE fragmentation, and that formation of F-actin shell, which is required for efficient NE fragmentation, may be disrupted at pH_i 6.7. In the experiments for revision, our staining showed that F-actin shell was not disturbed even when pH_i was clamped at 6.7. Lenart Lab proposed that small F-actin protrusions formed on the shell is essential for efficient NE fragmentation (Mori et al., 2014; Wesolowska et al., 2018 *preprint*). Thus, protrusions on the shell may be disrupted at pH_i of 6.7. Unfortunately, the protrusions were not visible on our imaging probably due to insufficient resolution. Thus we have added description about protrusions in Discussion as an issue to be investigated in future study (pages 16, lines 369-376).

Lenart Lab described actin meshwork formation and subsequent chromosome transport as events that occur 'after GVBD'. We follow their papers and describe these events as those 'after GVBD' in our revised manuscript. However, we had used a term of "completion of GVBD" to refer to completion of cytoplasmic granule invasion. This is definitely confusing for readers because the granule invasion proceeds during a period of actin meshwork-dependent chromosome transport (chromosome transport 'after GVBD' occurs during a period in which granule invasion proceeds toward 'completion of GVBD'). Thus we decided not to use 'completion of GVBD'. Instead, we described GVBD as a moment at which the granule invasion starts, and progression of the granule invasion as an event after GVBD (chromosome transport and granule invasion proceed after GVBD) in the revised manuscript.

On the basis of these observations, we have revised a model of SGK-dependent

meiotic resumption in ovarian oocytes. So far it is still unclear whether incompleting granule invasion causes serious consequences in oocyte maturation and/or following development, our finding of defects in chromosome gathering and spindle formation at a low pH_i further emphasizes importance of pH_i increase in ovarian oocytes. So, in the revised model, we describe that sfSGK-dependent pH_i increase is a prerequisite for actin-driven chromosome transport and microtubule organization for spindle formation in ovarian oocytes. We have added Fig 8 and description about this model (page 15, line 349-356).

Bun et al. 2018. A disassembly-driven mechanism explains F-actin-mediated chromosome transport in starfish oocytes. *eLife*. 7:e31469. doi:10.7554/eLife.31469.

Burdyniuk et al. 2018. F-Actin nucleated on chromosomes coordinates their capture by microtubules in oocyte meiosis. *J. Cell Biol.* 2018 217:2661-2674.

Lénárt et al. 2005. A contractile nuclear actin network drives chromosome congression in oocytes. *Nature*. 436:812-818.

Mori et al. 2011. Intracellular transport by an anchored homogeneously contracting F-actin meshwork. *Curr. Biol.* 21:606-611.

Mori et al. 2014. An Arp2/3 nucleated F-actin shell fragments nuclear membranes at nuclear envelope breakdown in starfish oocytes. *Curr. Bio.* 24:1421–1428.

Wesolowska et al. 2018. An F-actin shell ruptures the nuclear envelope by sorting pore-dense and pore-free membranes in meiosis of starfish oocytes. *bioRxiv*. doi.org/10.1101/480434 (Preprint posted November 28, 2018).

5. Throughout the manuscript the format of citations should be revised.

We have revised the format of all citations according to author instructions of JCB.

6. Line 23. fertilizability?

Fertilizability means capability of being fertilized. We have revised the sentence from “oocytes resume meiosis to acquire fertilizability” to “oocytes resume meiosis to become fertilizable” (pages 2, lane 35).

Reviewer #3

(Comments to the Authors (Required)):

Overall appreciation

The article by Hosada et al explores the role played by SGK (serum and glucocorticoid-regulated kinase) during oocyte maturation in starfish, rather than Akt/PKB pathway as previously reported. The authors convincingly demonstrate that following 1-methyl adenine induced maturation, that SGK is activated via a two-step process: first the C-terminal HM motif is phosphorylated by TORC2A-loop allowing interaction between SGK and PDK1 which phosphorylates the A-loop causing SGK activation. Both PDK1 and TORC2 activation require PI3 kinase upstream activity. This causes germinal vesicle breakdown and is associated with two separate events: a pH increase from pH6.7 (the pH inside starfish ovary) to pH7.2 (in sea water) or pH7.0 (if the oocyte remains in the ovary) and activation of Cdk1 via SGK-induced phosphorylation of Cdc25 (activating phosphorylation) and Myt1 (inhibitory phosphorylation). Interestingly, preventing the pH increase delays GV breakdown but not the activation of Cdk1. Furthermore, the pH increase is required for the migration of granules into the GV space following breakdown. Finally, since SGK activation and phosphorylation on the A loop peaks then reduces following GV breakdown, an unidentified SGK A loop phosphatase is proposed to play a crucial role in limiting the pH rise in oocytes within the ovary to pH7.0: this may prolong meiosis I thereby ensuring monospermy. This is a very interesting, thorough and convincing demonstration of the molecular events leading to oocyte maturation by a team that has led the way in cell cycle regulation of oocyte maturation. I am very positive about this manuscript.

Detailed summary of results and comments

Reagents

Antibodies generated to starfish SGK3:

ΔN50 Ab to SGK lacking N-ter 50aa (retains Thr312 PDK1 phosph motif in A loop)

HM Ab to SGK 17aa C-ter peptide containing hydrophobic motif (site of Ser479 mTORC2 phosphorylation)

Both antibodies detected 56kDa protein, and detected mobility shift after IMA

sfSGK-p312 A-loop Ab: Anti Phospho Human antibody to A loop phospho SGK: band detected in IMA oocytes but not before IMA. Band also 56kDa

Confirmed cross reactivity of HsAb by using purified starfish SGK and probing with HsAb.

All following figures strongly support the scientific claims.

Fig. 1. Sequence alignment and antibody cross-reactivity.

Fig. 2. Kinetics: After IMA sfSGK is activated 1 min., Cdc25 7 min., Cdk1 at 10 min. and GVBD 17 min. in isolated oocytes

Mobility shift and A loop phospho. detected 1 min. after IMA, peaked at 3 min.

Cdc25 hyper phosph. at 7 min. (Fig. 2a)

Cdk1 Tyr15 dephosph. at 10 min. (Fig. 2a)

- A loop Ab used for Thr312 of SGK
- Checked Ser477, of Akt by TORC2 phospho. Similar to SGK
- Checked ovary epithelium, no antibody signal detected (Fig. 2b)

In situ kinetics of ovary intact oocytes by injecting IMA into body cavity (similar to isolated oocytes)

GVBD in 25 min. after IMA (spawning at 30 min.) (Fig. 2c)

SGK and Cdc25 activated within 5 min, Cdk1 within 25 min.

Involvement of PDK1 and TORC2

A loop phosph Thr312: inhibited by BX795 (PDK1 inh). (Fig. 2d)

HM phosph Ser479: inhibited by pp242 (a TOR catalytic subunit inh). pp242 also inh. A loop phosph (cryptic site) (Fig. 2d)

Mobility shift: inhibited by pp242 but not BX795. (Fig. 2e)

Thus, shift caused by HM phosph. by TORC2.

Rapamycin had no effect on phospho, thus ruling out TORC1 (which also contains TOR cat. subunit) (Fig. 2e)

Mobility shift indicated that upper migrating band was the active dual phosphorylated form.

Lower band = non phospho

Middle band = Ser479, TORC2 phospho band (pp242 inh)

Upper band = Thr312, PDK1 and Ser479, TORC2 phospho band: Fig. 2e

First, only upper band was absent following BX which targets Thr312. Fig. 2d, lane 10. Fig. 2f, lane 5.

Second, upper and middle bands absent following pp242 which targets Thr479, Fig. 2d, lane 6. Fig. 2f, lane 6.

Third, upper and middle band were absent following wortmanin (PI3K inh). Fig. 2d, lane 4. Fig. 2f, lane 4. Some residual Phosph. Of middle band.
Mobility shift not inhibited by BX, Fig.2d, lane 10.
Thus, TORC2 Thr479 phospho. necessary for PDK1 Ser312 phospho. Both require PI3K

Fig. 3. SGK activation necessary for pH increase

Anti-SGK HM blocked mobility shift of SGK following microinjection and 1MA - Fig. 3a
Anti-SGK HM also blocked pH increase

Fig. 4. What causes Cdk1 activation - pH increase or SGK activation?

Previously GVBD occurred even when pH increase was blocked.
However, Anti-SGK HM blocked GVBD
Anti-SGK HM blocked Cdc25 activation and Cdk1 Tyr dephosph/activation. Fig. 4c.
Rescue of Anti-SGK HM with mRNA encoding SGK-T479E, 1MA now caused Cdk1 activation and GVBD - Fig. 4
Also, the pH increase and Cdk1 activation are independent events.

Fig. 5. Reduced pH increase delays GVBD and granule invasion of GV but does not affect time of Cdk1 activation.

Cdk1 activation kinetics independent of pH increase.

Supp. Fig. 1. pH increase following 1 MeAd is not blocked by Cdk1 kinase inhibitor roscovitine

Supp. Fig. 2. GV envelope morphology at different pHs.

Supp. Fig. 3. Low pH delays complete invasion of granules into GV space following breakdown.

Movie 1. Beautiful GVBD in ASW. Granule invasion complete at 30min, but scale bar is missing.

Movie 2. Beautiful GVBD with reduced granule invasion (at 1hr20min) at pH6.7, but scale bar is missing.

Movie 3. Beautiful GVBD with reduced granule invasion (at 1hr10min) at pH7.0, but scale bar is missing.

Movie 4. Beautiful GVBD with granule invasion at pH7.2 (at 40-50min), but scale bar is missing. We have added scale bars to all of the movies (see Videos 1-4).

Additional comments

Interesting to note that the pH increase induced by SGK is damped and remains at pH7.0 so that oocytes enter MI arrest induced by pH7.0, thus leading to monospermy. This is associated with dephosphorylation of the A loop by an as yet unknown phosphatase following Cdk1 activation (Fig. 5a)

It is also interesting to note that SGK phosphorylates Cdc25 (activating) and Myt1 (inhibiting) thus promoting Cdk1 Tyr 15 dephosphorylation (activation).

We would like to inform about a change in definition of GVBD in the revised manuscript. In response to comments from reviewer 1 and 2, we performed fluorescent imaging of fixed oocytes. Our interpretations of the results are based on a series of excellent works involving function of actin architectures in starfish oocytes by Lenart Lab. They have described formation of some F-actin architectures and actin-dependent chromosome transport occur 'after GVBD'. We follow their papers, and describe the F-actin dependent transport as an event 'after GVBD' in our revised manuscript. However, we had used a term of "completion of GVBD" to refer to completion of cytoplasmic granule invasion. This is definitely confusing for readers because the granule invasion proceeds during a period of the actin-dependent chromosome transport (chromosome transport 'after GVBD' occurs during a period in which granule invasion proceeds toward 'completion of GVBD'). Thus we decided not to use 'completion of GVBD'. Instead, we describe GVBD as a moment at which the granule invasion starts, and progression of the granule invasion as an event after GVBD.

August 7, 2019

RE: JCB Manuscript #201812133R

Prof. Kazuyoshi Chiba
Ochanomizu University
Department of Biological Sciences
2-1-1 Ohtsuka,
Bunkyo-ku,, Tokyo 112-8610
Japan

Dear Prof. Chiba:

Thank you for submitting your revised manuscript entitled "SGK regulates pH increase and cyclin B-Cdk1 activation to resume meiosis in starfish ovarian oocytes". The paper has now been seen again by reviewer #1 who now recommends publication, pending some final minor (textual) revisions. Therefore, we would be happy to publish your paper in JCB pending final revisions necessary to meet our formatting guidelines (see details below). Please also be sure to address reviewer #1's final comments.

A. MANUSCRIPT ORGANIZATION AND FORMATTING:

Full guidelines are available on our Instructions for Authors page, <http://jcb.rupress.org/submission-guidelines#revised>. **Submission of a paper that does not conform to JCB guidelines will delay the acceptance of your manuscript.**

1) Text limits: Character count for Articles and Tools is < 40,000, not including spaces. Count includes title page, abstract, introduction, results, discussion, acknowledgments, and figure legends. Count does not include materials and methods, references, tables, or supplemental legends. You currently meet this limit but please bear it in mind when revising.

2) Figure formatting: Scale bars must be present on all microscopy images, including inset magnifications. Molecular weight or nucleic acid size markers must be included on all gel electrophoresis.

3) Statistical analysis: Error bars on graphic representations of numerical data must be clearly described in the figure legend. The number of independent data points (n) represented in a graph must be indicated in the legend. Statistical methods should be explained in full in the materials and methods. For figures presenting pooled data the statistical measure should be defined in the figure legends. Please also be sure to indicate the statistical tests used in each of your experiments (both in the figure legend itself and in a separate methods section) as well as the parameters of the test (for example, if you ran a t-test, please indicate if it was one- or two-sided, etc.). Also, if you used parametric tests, please indicate if the data distribution was tested for normality (and if so, how). If not, you must state something to the effect that "Data distribution was assumed to be normal but

this was not formally tested."

4) Title: The title should be less than 100 characters including spaces. Make the title concise but accessible to a general readership. While we largely agree with your title, we do not feel that the word "ovarian" is necessary so we recommend removing it.

5) Materials and methods: Should be comprehensive and not simply reference a previous publication for details on how an experiment was performed. Please provide full descriptions (at least in brief) in the text for readers who may not have access to referenced manuscripts.

6) Please be sure to provide the sequences for all of your primers/oligos and RNAi constructs in the materials and methods. You must also indicate in the methods the source, species, and catalog numbers (where appropriate) for all of your antibodies.

7) Microscope image acquisition: The following information must be provided about the acquisition and processing of images:

a. Make and model of microscope

b. Type, magnification, and numerical aperture of the objective lenses

c. Temperature

d. imaging medium

e. Fluorochromes

f. Camera make and model

g. Acquisition software

h. Any software used for image processing subsequent to data acquisition. Please include details and types of operations involved (e.g., type of deconvolution, 3D reconstitutions, surface or volume rendering, gamma adjustments, etc.).

8) References: There is no limit to the number of references cited in a manuscript. References should be cited parenthetically in the text by author and year of publication. Abbreviate the names of journals according to PubMed.

9) Supplemental materials: There are strict limits on the allowable amount of supplemental data. Articles/Tools may have up to 5 supplemental figures. You currently meet this limit but please bear it in mind when revising.

Please also note that tables, like figures, should be provided as individual, editable files. A summary of all supplemental material should appear at the end of the Materials and methods section.

10) Conflict of interest statement: JCB requires inclusion of a statement in the acknowledgements regarding competing financial interests. If no competing financial interests exist, please include the following statement: "The authors declare no competing financial interests." If competing interests are declared, please follow your statement of these competing interests with the following statement: "The authors declare no further competing financial interests."

11) ORCID IDs: ORCID IDs are unique identifiers allowing researchers to create a record of their various scholarly contributions in a single place. At resubmission of your final files, please consider providing an ORCID ID for as many contributing authors as possible.

B. FINAL FILES:

-- High-resolution figure and video files: See our detailed guidelines for preparing your production-ready images, <http://jcb.rupress.org/fig-vid-guidelines>.

Thank you for this interesting contribution, we look forward to publishing your paper in Journal of Cell Biology.

Sincerely,

Jan Ellenberg, PhD
Monitoring Editor
JCB

Tim Spencer, PhD
Deputy Editor
Journal of Cell Biology

Reviewer #1 (Comments to the Authors (Required)):

I am impressed by the revised version of the manuscript and I recommend publication without further changes. The authors fully answered all my previous concerns, and the additional immunofluorescence data clarifying the effects of intracellular pH changes significantly improved the manuscript. The only point is that in my view, the major defect caused by changed pH is on microtubule dynamics and spindle assembly. I would suggest to emphasize this and in particular I

would suggest to change the final scheme accordingly (effects on actin-driven chromosome congression and invasion of yolk vesicles may be secondary effects of perturbed spindle assembly). Furthermore, it could be discussed more clearly whether the authors think that these effects on spindle assembly are direct effects of pH, or are they a consequence of delayed activation of cdk1.

JCB manuscript #201812133R

(Hosoda et al., SGK regulates pH increase and cyclin B-Cdk1 activation to resume meiosis in starfish ovarian oocytes)

Response to reviewers' comments and suggestions

First of all, we would like to thank reviewer#1 for his/her further suggestions. Below, the review comments are pasted with blue, and our responses are described with black. We hope that these responses meet the suggestions.

Reviewer #1

(Comments to the Authors (Required)):

I am impressed by the revised version of the manuscript and I recommend publication without further changes. The authors fully answered all my previous concerns, and the additional immunofluorescence data clarifying the effects of intracellular pH changes significantly improved the manuscript. The only point is that in my view, the major defect caused by changed pH is on microtubule dynamics and spindle assembly. I would suggest to emphasize this and in particular I would suggest to change the final scheme accordingly (effects on actin-driven chromosome congression and invasion of yolk vesicles may be secondary effects of perturbed spindle assembly).

According to the suggestions, we revised our manuscript to emphasize the defects on microtubule organization. In Results in our manuscript of the 1st revision, we had described defects on chromosome transport and microtubule organization in a paragraph. To emphasize the latter, we have separated the description about the microtubules at low pH_i from the paragraph (pages 14, lines 316-330), and described it as a new independent paragraph beginning with a newly added sentence “The most remarkable defect at a clamped pH_i of 6.7 was found on microtubule organization” (page 14, line 323).

Similarly, we have revised the last sentence in the first paragraph in Discussion from “In particular, actin-dependent chromosome transport and microtubule organization for spindle formation are defective at pH_i 6.7” to “In particular, actin-dependent chromosome transport is less efficient and microtubule organization for spindle assembly is severely disturbed at pH_i 6.7” (page 15, lines 340-341).

We have also revised the final scheme in Figure 8. As reviewer#1 suggested, the effects on actin-driven chromosome transport and invasion of cytoplasmic granules may be secondary effects of perturbed spindle assembly. However, we think that this is less plausible because disruption of microtubule dynamics by nocodazole does not prevent actin-dependent transport (Lénárt et al., 2005). So far, it remains still unclear whether there are causal links among defects on actin-driven chromosome transport, microtubule organization and granule invasion. In this context, our scheme in Figure 8 was somewhat misleading because in the figure, the blue arrow from “High $pH_i \sim 7.0$ ” pointed to only “F-actin-driven chromosome transport” and “cytoplasmic granule invasion”. In addition, these two events and “spindle assembly” were linked by an arrow. Therefore, when readers look at this scheme, they may feel that $pH_i 7.0$ is required for “chromosome transport” and “granule invasion”, and that these two events lead to microtubule organization that enables spindle assembly. In the revised scheme, we have added “microtubule organization” side-by-side with “F-actin-driven chromosome transport” to show $pH_i 7.0$ is also required for microtubule organization. The following are the revised figure and its old version.

Old figure

Revised figure

In addition, although we had used terms of “spindle formation” and “spindle assembly” in the main text, we replaced “spindle formation” with “spindle assembly” in the present revision, because both terms have the same meaning and “assembly” is more generally used.

Lénárt et al. 2005. A contractile nuclear actin network drives chromosome congression in oocytes. *Nature*. 436:812-818.

Furthermore, it could be discussed more clearly whether the authors think that these effects on spindle assembly are direct effects of pH_i, or are they a consequence of delayed activation of cdk1.

We think that the defects on spindle assembly is not due to the effects of pH_i changes on cyclin B-Cdk1, because cyclin B-Cdk1 was activated several minutes earlier at low pH_i than at high pH_i (Fig. 5A, pages 12, lines 267-269), and its activity was maintained at least during our observation of the defects (Fig. S2, pages 12, lines 281-282). Rather, we speculate that the defects are direct effects of pH. Accordingly, we have revised a paragraph about defects of microtubules in Discussion (pages 17, lines 387-390) as follows.

Old paragraph

Microtubule organization was also sensitive to pH_i changes (Fig. 7). This finding is consistent with previous studies showing that the pH can affect microtubule polymerization/depolymerization *in vitro* (Regula et al., 1981) and microtubule organization *in vivo* in green alga (Liu et al., 2017) and sea urchin eggs (Harris and Clason, 1992). Furthermore, the mitotic spindles were poorly organized at pH_i 6.3 in fertilized sand dollar eggs (Watanabe et al., 1997). Thus, defective spindle assembly at reduced pH_i values seems to be a common feature in meiosis and mitosis.

Revised paragraph

Microtubule organization was severely disturbed at a low pH_i (Fig. 7). We speculate that this defect is a consequence of direct effects of pH on microtubule dynamics because pH affects microtubule polymerization/depolymerization *in vitro* (Regula et al., 1981). pH can also affect microtubule organization *in vivo* in green alga (Liu et al., 2017) and sea urchin eggs (Harris and Clason, 1992). Furthermore, the mitotic spindles were poorly organized at pH_i 6.3 in fertilized sand dollar eggs (Watanabe et al., 1997). Thus, defective spindle assembly at reduced pH_i values seems to be a common feature in meiosis and mitosis.